# IκBα controls dormancy in hematopoietic stem cells via retinoic acid during embryonic development

Roshana Thambyrajah [1,2,8] ✉, Maria Maqueda[1,2,3], Muhammad Zaki Fadlullah [4], Martin Proffitt[1], Wen Hao Neo [4], Yolanda Guillén[1], Marta Casado-Pelaez [2], Patricia Herrero-Molinero[1], Carla Brujas[1], Noemi Castelluccio [1,7], Jessica González [1,2,3], Arnau Iglesias[1,2,3], Laura Marruecos[1], Cristina Ruiz-Herguido[1], Manel Esteller [2,3,5,6], Elisabetta Mereu [2], Georges Lacaud [4], Lluis Espinosa [1,3] & Anna Bigas [1,2,3,8] ✉

Recent findings suggest that Hematopoietic Stem Cells (HSC) and progenitors arise simultaneously and independently of each other already in the embryonic aorta-gonad mesonephros region, but it is still unknown how their different features are established. Here, we uncover IκBα (*Nfkbia*, the inhibitor of NF-κB) as a critical regulator of HSC proliferation throughout development. IκBα balances retinoic acid signaling levels together with the epigenetic silencer, PRC2, specifically in HSCs. Loss of *IκBα* decreases proliferation of HSC and induces a dormancy related gene expression signature instead. Also, IκBα deficient HSCs respond with superior activation to in vitro culture and in serial transplantation. At the molecular level, chromatin regions harboring binding motifs for retinoic acid signaling are hypo-methylated for the PRC2 dependent H3K27me3 mark in IκBα deficient HSCs. Overall, we show that the proliferation index in the developing HSCs is regulated by a IκBα-PRC2 axis, which controls retinoic acid signaling.

HSCs can replenish the adult blood system by generating mature blood cells of all lineages through intermediate stages of multipotent progenitors[1]. Understanding the molecular programming underlying the formation of HSCs during development is critical to improve the feasibility of generating them from Pluripotent Stem cells, or through re-programming of somatic cells to treat blood malignancies[2]. Blood stem and progenitor cell (HSPC) emerge through trans-differentiation of specialized endothelial-like cells, termed hemogenic endothelium (HE), and the first cells capable to regenerate the hematopoietic

system are found in the Aorta-Gonad-Mesonephros (AGM) region between embryonic day (E) 10.25 and 11.5 in the mouse embryo[3–5] and accumulate as intra-aortic hematopoietic clusters (IAHC) within the ventral wall of the dorsal aorta (vDA, DA)[6–8].

Hemogenic potential in the DA can be readily identified by the expression of transcription factors such as *Gata2* and *Gfi1*[9–12] and a few cells in IAHC co-express markers associated with HSC activity. Some of the IAHC that are cKIT and CD41 positive (Pre-HSCs/T1 HSCs), gain HSC-specific markers, including CD45, SCA1, and EPCR (CD201/

[1]Program in Cancer Research, Hospital del Mar Research Institute, Barcelona, Spain. [2]Josep Carreras Leukemia Research Institute, Barcelona, Spain. [3]Centro de Investigacion Biomedica en Red Cancer (CIBERONC), Madrid, Spain. [4]Cancer Research UK Stem Cell Biology Group, Cancer Research UK Manchester Institute, The University of Manchester, Manchester, UK. [5]Institucio Catalana de Recerca i Estudis Avançats (ICREA), Barcelona, Catalonia, Spain. [6]Physiological Sciences Department, School of Medicine and Health Sciences, University of Barcelona (UB), Barcelona, Catalonia, Spain. [7]Present address: Ghent University Hospital, Ghent, Belgium. [8]These authors jointly supervised this work: Roshana Thambyrajah, Anna Bigas. ✉e-mail: roshanathambyrajah@gmail.com; abigas@imim.es

PROCR) (T2 HSCs)[13]. From the AGM, the first HSCs migrate to the FL by E12.5 to amplify and acquire an adult HSC phenotype[14]. There, the hematopoietic hierarchy is first evident, and a pool of long-term HSCs (LT-HSCs, LSKCD48-CD150+) resides at its apex.

Recent studies indicate that (LT-) HSC and HPCs fate is already segregated in the AGM and that the fetal liver merely serves as a niche for their amplification as separate populations, with the (LT)-HSCs pool remaining smaller than the extensively proliferating HPC pool[15–17]. Signals that allow this segregation, and how these LT-HSCs protect their integrity, would help to understand this process. Studies in the adult bone marrow describe two subtypes of LT-HSCs, a small pool of LT-HSCs is classified as dormant, the most naïve state of HSCs, whilst the rest of the LT-HSCs are active and ready for cycling. The dormant LT-HSCs divide slower, have a lower metabolic rate, and respond with a delayed repopulation/activation capacity when transplanted, although they possess the most potent capability to regenerate the hematopoietic system under stress conditions.

However, it is unknown how, and at which developmental stage(s) this dormant state is induced. The main known regulator that acts as a switch between these states in the adult bone marrow is retinoic acid (RA) signaling[18,19], next to IFNα[20–22]. Intriguingly, RA has also been linked to HSC activity in the AGM, and LT-HSC mobilization in the adult bone marrow[23–26], indicating that signals that regulate the pool of inactive/dormant HSC in the adult might already be established in the AGM. Moreover, several inflammatory cytokines contribute to HSPC formation in the AGM[27] and often convene on NF-κB activation[28–30].

In this context, IκBα (*Nfkbia*) has a well-established role as a repressor of NF-κB factors such as p65 (*RelA*) and p50 (*Nfkb1*) mainly in the cytoplasm, but also as a terminator of the signal activation by escorting p65 out of the nucleus[31]. Yet, IκBα itself can exert gene regulation by direct interaction with chromatin. Chromatin-bound IκBα regulates stem cell-specific genes in association with elements of the polycomb repression complex 2 (PRC2), which catalyzes the histone mark H3K27me3, at specific genetic loci, e.g., in intestinal and germline stem cells[32–34].

Here, we uncover a specific reduction of LT-HSCs in the *IκBα* knockout newborn pups. This defect is manifested at the FL stage already, and traceable to the first emerging HSCs in the E11.5 AGM, without affecting the general HPC population. We present data indicating that IκBα exerts a PRC2-dependent function in this system, and specifically regulates the RA signaling pathway. In support of our findings, we find IκBα binding to the promoter of *Rara* and *Rary,* and their gene expression levels are up-regulated upon IκBα loss in purified LT-HSCs. In parallel, we detect decreased levels of the PRC2-dependent H3K27me3 mark at discreet loci that enrich for the RARα, RXRα, and RXRγ motifs, further suggesting that loss of IκBα -PRC2 interactions widely affects RA signaling in *IκBα* KO LT-HSCs. In addition, the *IκBα* KO LT-HSCs manifest a dormancy signature at the molecular level, and functionally, they display a slow-cycling and dormant behavior in Ki67 and BrdU labeling experiments. The dormancy phenotype is also evident in transplantation assays, where recipients of purified *IκBα* KO LT-HSCs show a delay to reach blood chimerism to WT levels. In fact, consistent with the re-mobilization of dormant cells, *IκBα* KO LT-HSCs outperform WT LT-HSCs in terms of blood chimerism in secondary and tertiary recipients.

Overall, we identify a nuclear IκBα-PRC2 interactions as a regulatory axis controlling the balance between HSC proliferation and dormancy, starting from the AGM stage of HSC ontogeny.

## Results

### NF-κB signaling is significantly enriched throughout HSC ontogeny

To understand the precise involvement of the inflammatory signaling pathways throughout HSC ontogeny, we scrutinized a publicly available dataset of single-cell sequencing of HSCs at various stages of development, i.e., AGM, FL, and bone marrow[13]. Cells clustered according to their cell fate, with AGM endothelial cells (EC) situated at the opposing end to the bone marrow HSCs (Fig. 1A). In between these two fates, the T1/T2 HSC of the AGM, and the FL HSCs (E12.5 and E14.5) formed a continuum towards the bone marrow HSCs (Fig. 1A).

We assessed these different HSC populations specifically for gene expression levels of inflammatory signaling and NF-κB signatures (Fig. 1B, Supplementary Data 1), and found its enrichment to increase from early HSC stages to bone marrow LT-HSCs, with a subtle decrease at fetal stages. Altogether, these findings indicate a dynamic involvement of inflammatory pathways from (pre-) HSC stages to phenotypically adult HSC development.

We next compared the status of NF-κB signaling activity at the early developmental stages by performing Gene Set Enrichment Analysis (GSEA). Ranked T1/T2 HSC vs E12/14.5 fetal liver (FL-HSC) and EC vs T1/T2 HSC differentially expressed genes significantly correlated to the hallmark "Tnfa signaling via NF-κB" (Fig. 1C and Supplementary Data 1). Subsequently, we examined the expression profile of specific NF-κB signaling elements across identified cell populations (Fig. 1D, Supplementary Fig. S1) finding more cells in the adult bone marrow HSCs showing higher expression of these genes. Nevertheless, some cells in the T1/T2 and FL-HSC populations also expressed NF-κB signaling elements *RelA*, *Rel*, *Nfkbia*, and Nfkbie, supporting a role for NF-κB in mouse HSC development, as previously shown for HSPC generation in zebrafish[30].

### The number of LT-HSCs is reduced in the hematopoietic sites of *IκBα* KO

To examine the putative effect of NF-κB over-activation in HSC development, we took advantage of the *IκBα* KO mice which show normal embryonic development but then succumb to death beyond 7 days after birth due to massive inflammation[35,36]. We analyzed the stem cell compartment of *IκBα* deficient mice at early postnatal stages (postnatal days 5 and 6, P5/6) in the liver and bone marrow and FL at E14.5 (Fig. 2A).

All genotypes showed comparable numbers of cellularity in the bone marrow and (fetal/newborn) liver (Supplementary Fig. S1B), and all blood lineages were present in the *IκBα* KO newborn, albeit mild but significant lineage skewing (Supplementary Fig. S1C), as previously reported[37,38]. The frequency of the LSK was comparable between the genotypes in the E14.5 FL, and the newborn liver and bone marrow (Fig. 2i, Ci and Supplementary Fig. S2A). However, the number of LT-HSCs was significantly reduced in newborn BM and E14.5 fetal liver of *IκBα* KO (Fig. 2Bii, Cii and Supplementary Fig. S2A and B).

Since the loss of the NF-κB signaling downstream effector, p65, in mice is embryonic lethal at 15-16 days of gestation and is characterized by an enormous degeneration of the fetal liver by apoptosis[39], we evaluated the cellularity, and the NF-κB signaling activity status of the FL in *IκBα* KO at E16.5. We did not detect any significant alterations in cell number, appearance, or weight between the genotypes (Supplementary Fig. S1B and S3A). Additionally, we checked the NF-κB activity status by IHC for p65 in *IκBα* deficient FL and detected lower numbers of nuclear p65 in *IκBα* KO (Supplementary Fig. S3B, C). These findings are in line with the initial study of the *IκBα* KO phenotype, where the authors found no, or variable levels of NF-κB over-activation in different cell types, due to the IκBα function as NF-κB inhibitor being compensated by IκBβ[39].

Occasionally, *IκBα* KO mice survive to adulthood for unexplained reasons, and even in these rare survivors, we detected a reduction in LT- HSCs numbers (Supplementary Fig. S3D). Finally, we did not find any statistically significant changes between *IκBα* WT and HET regarding contribution to the LSK and LT-HSC population (Fig. 2B, C).

### FL LT-HSCs retain an AGM-specific gene expression profile in *IκBα* KO

To define the molecular alterations in the stem cell compartment in *IκBα* KO, we followed two complementary strategies and used the E14.5

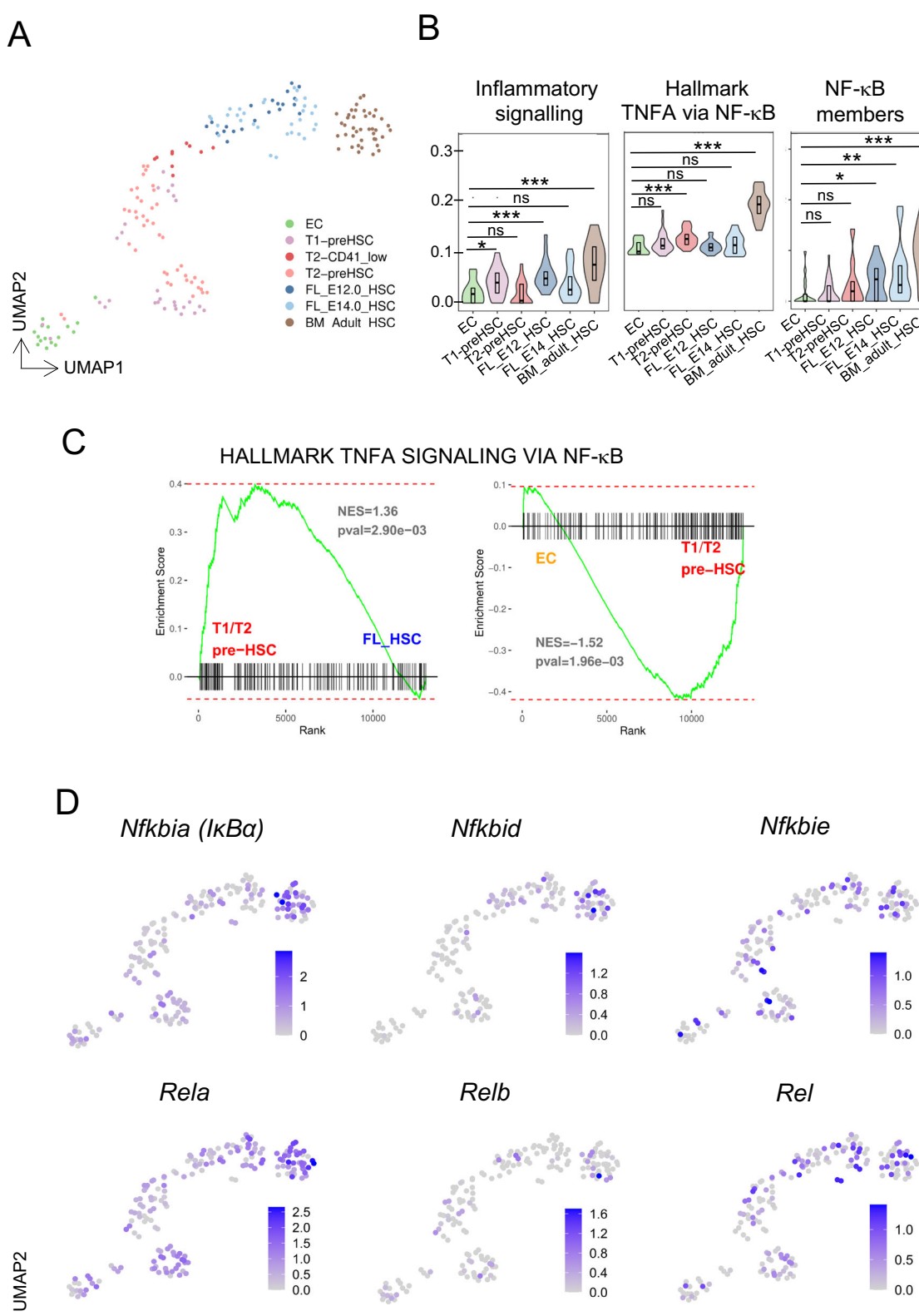

FL as the source for the cells. We either sorted 5000 LSK cells from *IκBα* WT or KO for single-cell RNA-sequencing to estimate alterations across different (stem) blood lineages, and in parallel, we purified 500 WT, HET or *IκBα* KO LT-HSCs (LSK/CD150+CD48-) by FACS to perform bulk RNA-seq for detailed transcriptomic analysis of the samples (Fig. 2D, Supplementary Data 2 and 3). Within the single-cell RNA-seq data set of the LSK, we assigned cell identity based on previous studies

(Supplementary Fig. S4A–C, Supplementary Data T2)[40] and confirmed the cell identity assignment by ordering the cells in a pseudo-time trajectory based on their overall transcriptomic profile (Supplementary Fig. S4A–C).

Both, the *IκBα* WT and KO cells contributed to all hematopoietic stem and progenitor populations (Supplementary Fig. S4D). We performed differentially expressed genes (DEG) analysis for all the sub-

**Fig. 1 | NF-κB signaling is enriched in AGM-derived HSCs. A** UMAP projection colored by scRNAseq clusters where different HSC populations are highlighted. Single-cell RNA-sequencing data derived from Zhou et al.[13]. EC = CD31+Cdh5+CD41-CD43-CD45-; T1 PreHSC = CD31+cKIT+CD41[low]CD201[high]*CD45-*; T2 PreHSC = CD31+cKIT+CD201[high]*CD45+*; E12 FL-HSC= lin-SCA+CD201[high]Mac1[low]; E14 FL-HSC = CD45+CD201CD48-CD150+; adult BM-HSC= lin-SCA1+cKIT+CD135-CD34- CD48-CD150+ (**B**), enrichment score analysis of indicated HSC populations for inflammatory, NF-κB pathway and NF-κB signaling molecules. Boxplot shows indicate the upper- (75%) and lower- (25%) percentile, with the median value shown as a solid black line. Statistical significance determined with a two-sided *t*-test. No adjustments were made for multiple comparisons. *** < 0.001, ** < 0.01, * < 0.05 (**C**), GSEA results comparing T1/T2 HSC cells to FL-HSC (left) or endothelial cells (EC) to T1/T2 HSC for correlation to the Hallmark "TNFA_SIGNALING_VIA_ NF-κB" gene set. The indicated *p*-value is a nominal *p*-value obtained from GSEA. **D** UMAP projection colored by the expression of selected genes from the NF-κB signaling pathway.

populations of the LSK by comparing the levels of gene expression in the single cells between *IκBα* WT and KO and determined if a significant number of cells were consistently expressing a given gene at distinct levels (Supplementary Data 2). We then queried if the NF-κB pathway was over-activated in any of the stem cell/progenitor populations in the *IκBα* KO. Thus, we plotted the levels of gene expression for genes that are identified as NF-κB related (KEGG pathway ID mmu04064) across the entire LSK populations separated by their genotype, *IκBα* WT, and KO. Although in some instances, i.e., LT- and ST-HSCs, there might be a few more cells in *IκBα* KO that have higher levels of this signature, statistical tests rendered them insignificant (Fig. 2E and Supplementary Fig. S4E).

Next, we analyzed the transcriptional changes in the LT-HSC by bulk RNA-seq of purified LT-HSCs. Principal component analysis (PCA) showed samples grouping based on their genotype, particularly, PC1 (48% total variance) separated WT/HET and KO samples (Supplementary Fig. S4F and G). A total of 1476 differentially expressed genes between KO and *IκBα* WT/HET were detected (shrunken logFC > 1 FDR adjusted *p*-value < 0.05) with 92% of them being up-regulated (Fig. 3A, Supplementary Data 3). Interestingly, genes that were over-expressed in *IκBα* KO are genes involved in several inflammatory pathways, including "TnfA signaling via NFκB" (Fig. 2F). Nevertheless, a GSEA for the KEGG gene set of NF-κB (mmu04064) did not show a significant overall enrichment of this pathway in *IκBα* KO LT-HSCs (Fig. 2G). Moreover, we only found a few (16 out of 105) significantly DEG from this set in our RNA-seq data (Supplementary Fig. S5A). We reason that although elements of inflammatory signaling pathways are deregulated, this deregulation is not the main transcriptional change occurring in *IκBα* KO.

Instead, we recognized that several elements of the up-regulated pathways in *IκBα* KO LT-HSCs, including "Interferon alpha," "IL-6/JAK/ STAT", "Tgf-beta signaling", and "angiogenesis" (Fig. 2F and Supplementary Data 3) are critical for embryonic AGM-associated HSCs. In a previous study, we found that loss of *IκBα* caused retention of embryonic gene expression profile in adult intestinal cells[33]. We therefore examined the expression levels of key embryonic (AGM) HSC-associated genes such as *Smad6, Sox18, Gpr56 (Adgrg1), Gata2,* and *Neurl3* in our FL data set and found that all of them were significantly up-regulated in the *IκBα* KO FL HSCs (Fig. 3A and Supplementary Fig. S5B). These results suggested that *IκBα* deficiency affected the maturation of the HSCs from the AGM to the FL stage.

To evaluate this hypothesis further, we performed GSEA of genes up-regulated in *IκBα* KO LT-HSC against two data sets; one dataset consists of the top 300 genes up-regulated from hemogenic endothelium and HSCs single-cell RNA-seq and are therefore characteristic for nascent AGM HSCs[41]. The second dataset includes an exclusive FL LT-HSC signature[42]. Strikingly, the *IκBα* KO FL LT-HSC correlate at a much higher degree, and significantly with the AGM-associated HSC profile (NES = 2.02, *p*-value < 0.001), than with the FL LT-HSC signature (NES = 1.06, *p*-value > 0.05) (Fig. 3B, Supplementary Fig. S5B) indicating that *IκBα* KO FL LT-HSCs retain a gene signature of the AGM stage. Interestingly, in the significant down-regulated pathways, we find several processes, including MYC-targets, Oxidative phosphorylation, G2M checkpoint, and glycolysis in *IκBα* deficient LT-HSCs (Fig. 3C), reminiscent of quiescent HSCs and indicating a slow-cycling/dormant cell state.

## *IκBα* KO has reduced cKIT+CD45+SCA1+EPCR+HSC in the AGM

Since we had identified a retention of the AGM HSC signature in *IκBα* KO LT-HSCs of the E14.5 FL (Fig. 3B), and NF-κB activity is already required during AGM hematopoiesis[28–30] we aimed to study the spatial distribution of *IκBα* by performing Immunohistochemistry (IHC) on E11.5 AGM sagittal sections. We used the *Gfi1:tomato* embryos to enrich for HSC containing IAHC in the AGM[11].

Besides the expected cytoplasmic staining, we discovered an accumulation of a punctuated and nuclear signal for IκBα mainly in GFI1 positive HE and IAHC (Fig. 3D and Supplementary Fig. S5D). The nuclear localization of IκBα raised the possibility that we were additionally detecting the form of the (nuclear) IκBα that has been previously linked to stem cell function[33,34]. Phosphorylated, nuclear IκBα is protected from degradation by SUMOylation[43], and is predominantly detected as phosphorylated Ser32,36 IκBα with a specific antibody (p-IκBα). Thus, we examined the presence of p-IκBα in cKIT or GFI1 positive cells and detected nuclear p-IκBα staining mainly in IAHC (Fig. 3E and Supplementary Fig. S5E) with a heterogeneous distribution in around 50% cKIT positive cells (Fig. 3F), strongly suggesting this additional function for IκBα already in the IAHC of the AGM.

We next assessed the impact of *IκBα* deletion in E11.5 AGMs and compared *IκBα* WT and KO by FACS analysis. The frequency of the overall CD31+cKIT+/CD45+ (HSPCs) was not altered in *IκBα* KO embryos when compared with the controls (Fig. 3G(i), Supplementary Fig. S5F). However, when additional markers that restrict this population further to HSCs were included, i.e. by sub-gating for GFI1/EPCR or SCA1/EPCR[13,44,45] then we detected a significant decline in the percentage of HSCs in the *IκBα* KO (Fig. 3G(ii), Supplementary Fig. S5F and S6A, B). Importantly, classical NF-κB activity was not substantially increased in the IAHC/HSC after loss of *IκBα* as assessed by IHC for p65 and p65 Ser536 (activated form of p65) (Supplementary Fig. S6C–E).

In brief, we detect phosphorylated Ser32,36 IκBα in IAHC and find that the number of HSC precursors in the AGM is lower in *IκBα* KO. This difference in the number of LT-HSCs between WT and KO increases throughout hematopoietic development (Supplementary Fig. S5C), suggesting that the HSCs population in *IκBα* KO does not expand at the same rate as their WT counterpart.

## EZH2 and IκBα interact in IAHC of the AGM

To highlight the pathways that are over- or, under-represented in LT-HSCs of *IκBα* deficient fetal livers, we also assessed which transcription factors are likely to trigger the genes up-regulated in *IκBα* KO LT-HSCs. A GSEA against the ChEA database indicated that the up-regulated genes in *IκBα* KO LT-HSCs are enriched by several factors, but most notably by EZH2, EED, and BMI1, prominent members of the PRC2 and PRC1 complex (Fig. 4A and Supplementary Data 3). This finding further supported our initial hypothesis for a chromatin function of IκBα in HSCs which requires interaction with the PRC2 complex (Mulero et al.[34]). To demonstrate this interaction in IAHC, we applied proximity ligation assay (PLA), an antibody-based approach that detects interactions between two proteins of interest. We probed EZH2 and IκBα in this assay on E11.5 AGM sections. Remarkably, we spotted dotted signal, indicative for interactions between EZH2 and IκBα, in nuclei of sparse cells in IAHC (Fig. 4B and Supplementary Fig. S6F, G), adding more evidence for our postulation.

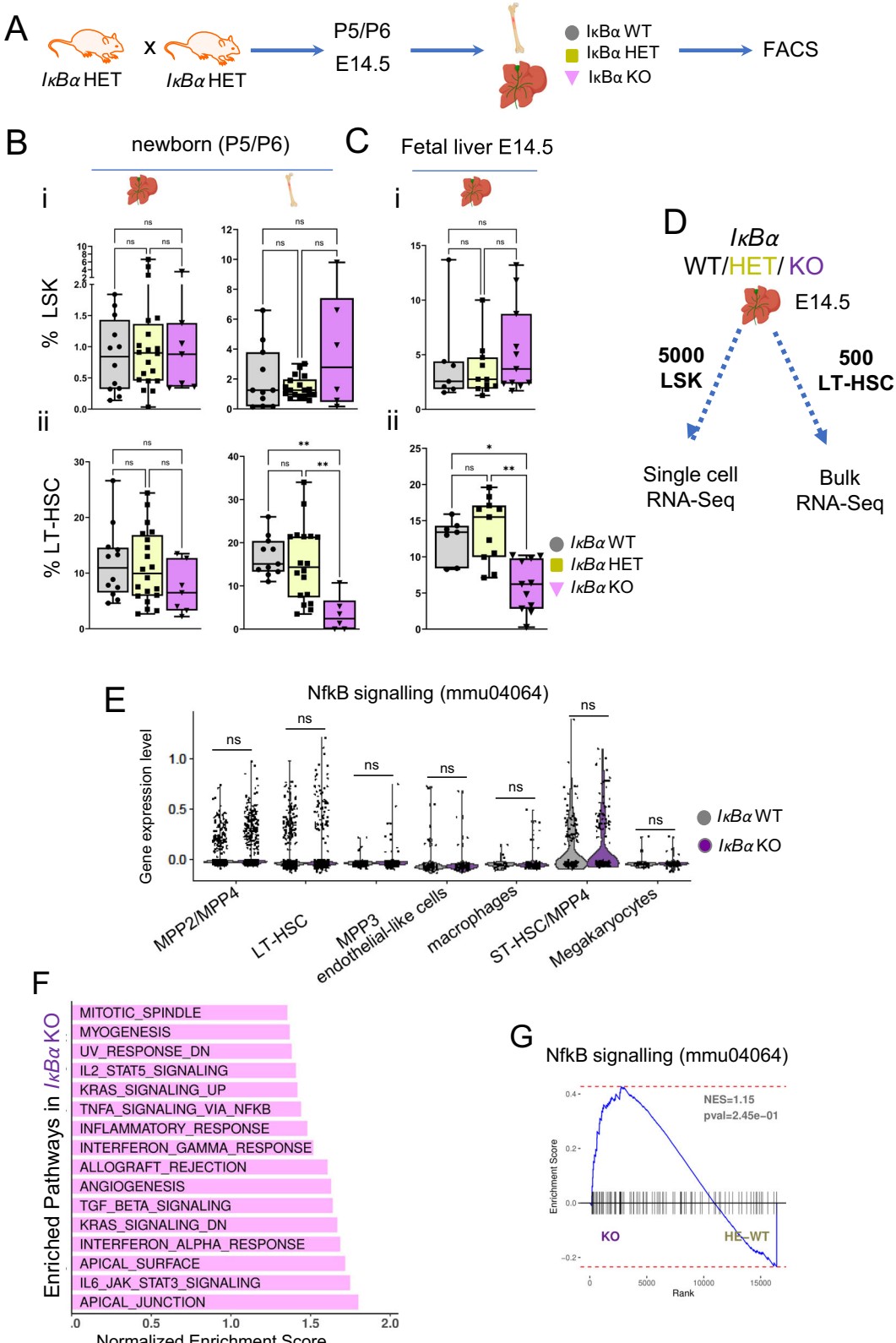

In brief, we have identified the interaction between PRC2 (EZH2) and IκBα in IAHC of the AGM.

**RA signaling is increased in the AGM and FL LT-HSCs of *IκBα* KO**
We then aimed to identify the loci that are regulated by IκBα and PRC2 binding. First, we performed CUT&Tag assay (C&T) with two different anti-IκBα antibodies targeting total IκBα (1st replicate in LT-HSC and

both replicates in CD31+) and p-IκBα Ser36 (2nd replicate in LT-HSCs) on sorted LT-HSC and isolated CD31 positive AGM cells (Fig. 4C). C&T assays with IκBα antibodies are challenging, but we obtained enrichment in specific regions with both antibodies to the promoter of the RA receptor, *Rarα*, in CD31+ cells from AGM, and at the promoter of the paralog, *Rarγ*, in FL LT-HSCs in both corresponding replicates (Fig. 4C). Intriguingly, both these receptors are over-expressed in the

**Fig. 2 | IκBα loss leads to decreased numbers of LT-HSCs at newborn and fetal stages. A** Scheme of the mouse mating to obtain *IκBα* WT, HET, and KO hematopoietic tissues. **B** Box plot with individual values of the frequency of (**i**) LSK (lin-SCA1+cKIT+) in the liver or bone marrow of newborn (P5/6) both obtained from *n* = 39 pups in 5 independent experiments and, (**ii**) LT-HSCs (LSKCD48-CD150+) from the previous gate (LSK) in the liver of newborn (P5/6) *IκBα* WT (*n* = 12), HET (*n* = 20) and KO (*n* = 7) and in the bone marrow of newborn (P5/6) *IκBα* WT (*n* = 11), HET (*n* = 18) and KO (*n* = 6). **C** Box plot with individual values of the frequency of (**i**) LSK in the fetal liver of E14.5 *IκBα* WT (*n* = 7), HET (*n* = 11) and KO (*n* = 11), and (**ii**) LT-HSCs (LSKCD48-CD150+) in E14.5 *IκBα* WT (*n* = 7), HET (*n* = 11) and KO (*n* = 11), both cased obtained from *n* = 19 embryos in 3 independent experiments. Statistical test for (**B**), (**C**): unpaired two-tailed Dunn's non-parametric all-pairs comparison test after Kruskal–Wallis test (**p-value < 0.01, *p-value < 0.05, ns p-value > 0.05). Box-plots show the median (center line) first and third quartiles (box limits), and whiskers extend to minimum and maximum values. **D** Scheme illustrating the two

strategies followed to explore the molecular changes in *IκBα* WT, HET, and KO. E14.5 FL were FACS sorted to obtain 5000 LSK cells for single-cell RNA-sequencing and 500 cells/sample of LT-HSCs for bulk RNA-sequencing. *n* = 4 embryos per genotype. **E** Violin plots with individual values depicting the expression levels from scRNAseq data of genes from NF-κB signaling (KEGG ID mmu04064) signature within each annotated cell population. Statistical test: two-tailed Wilcoxon rank sum test (ns *p*-value > 0.05). **F** Bar plot depicting significantly enriched MSigDB HALLMARK pathways in *IκBα* KO against WT/HET genotypes with positive Normalized Enrichment Score (NES), identified by GSEA from the bulk RNA-seq data (Benjamini–Hochberg procedure (FDR) adjusted p-value < 0.05). Bars sorted by NES. **G** NF-κB signaling signature (KEGG ID mmu04064) GSEA over the bulk RNA-seq data comparing *IκBα* WT/HET and KO LT-HSCs. Indicated *p*-value is a nominal *p*-value obtained from GSEA. Source data are provided as a Source Data file. **A**−**D** created with BioRender.com released under a Creative Commons Attribution-NonCommercial-NoDerivs 4.0 International license.

LT-HSCs of *IκBα* KO (Fig. 3A and Supplementary Data 3) and importantly, GSEA identified significant transcriptional enrichment for RARγ binding targets in LT-HSC of *IκBα* KO compared to WT/HET genotypes (Fig. 4A).

We further explored Rarγ activation by analyzing the expression levels of RARγ targets (TRANSFAC database ID M10353). We found a significant overlapping (hypergeometric test, *p*-value < 0.05) between the RARγ targets included in our RNA-seq dataset (2860 genes, Fig. 4D), and the identified DEGs (1476 genes), which were mostly up-regulated in the KO compared to WT/HET embryos (92%, Fig. 3A). Altogether these findings demonstrate an increased RARγ activation in the FL-HSC of the *IκBα* KO embryos.

To further understand the molecular consequences of *IκBα* deletion, we next analyzed the PRC2-dependent H3K27me3 status in *IκBα* WT and KO by C&T assay in FL HSCs. We detected decreased levels of H3K27me3 signal at the *Rarγ* promoter of *IκBα* KO LT-HSC (Fig. 4E) in two of out of three replicates. Additionally, 526 and 250 peaks were uniquely detected in the *IκBα* KO or WT LT-HSC, respectively (Supplementary Data 4). To uncover candidate transcription factors that were driving the epigenetic silencing of these regions in a IκBα-PRC2-dependent manner, we searched for *motifs* enriched in the unique H3K27 unmethylated regions of the *IκBα* KO LT-HSC (i.e., signal exclusively detected in 3 replicates of *IκBα* WT). Remarkably, one discovered motif was significantly similar to the RA receptors RXRγ, RxRα, and RARα known motifs (*q*-val < 0.05, HOCOMOCO core database) (Fig. 4F), suggesting that the IκBα-dependent PRC2 complex can regulate multiple elements of RA signaling in FL LT-HSCs. Therefore, up-regulation of RA targets could be due to IκBα or PRC2 loss of function.

We then examined if we could detect this perturbation of RA signaling already at the onset of HSC specification. Thus, we isolated CD31+cKIT- (endo) and CD31+cKIT+ IAHC from E11.5 AGMs of *IκBα* WT or KO and conducted quantitative PCR (qPCR) for RA signaling elements. Interestingly, *Rarγ* expression was significantly up-regulated in cKIT IAHC in *IκBα* KO AGMs (Fig. 4G).

In summary, we have uncovered an IκBα- PRC2 axis that specifically governs RA receptor signaling of RARα and RARγ in LT-HSCs and their precursors.

## Increased RA receptors in *IκBα* KO LT-HSCs impose a slow-cycling state

Since increased RA signaling is associated with dormancy of LT-HSCs in the bone marrow[18,46], and we detected changes in metabolic pathways that are associated with dormant cell state (Fig. 3C), we cross-examined the published list of key dormancy signature genes in LT-HSC with the RNA-seq data set of *IκBα* WT and KO LT-HSCs. Consistent with our assumption, key genes such as *p57*, *Tgm2*, *Rbp1*, *Cyp26b1* and *Ly6a* were up-regulated, while *Myc* and *Cdk6* showed lower expression in *IκBα* KO LT-HSCs, a pattern compatible with LT-HSC in a dormant

state (Fig. 5A). *IκBα* KO LT-HSCs also express higher levels of the transcription factor EB (*Tfeb*) (Fig. 5A and Supplementary Data 3), whose elevated levels is a further indicator of dormant cell[47,48].

Next, we investigated functionally whether *IκBα* deficient LT-HSCs present an RA-dependent dormant phenotype. We first determined their proliferation rate in vivo by injecting 2 mg of BrdU peritoneally into E13.5 pregnant females (Fig. 5B). After 2 h, the females were sacrificed to analyze the cell cycle status of FL LT-HSCs from WT and KO embryos by flow cytometry. We detected that a significantly lower proportion of LT-HSC had incorporated BrdU in the *IκBα* deficient embryos (Fig. 5B and Supplementary Fig. S7A), indicating a slower cycling of these cells. Next, we determined Ki67 levels by flow cytometry and detected significantly lower Ki67+ ST- and LT-HSCs in E14.5 FL HSCs of *IκBα* KO embryos (Fig. 5C and Supplementary Fig. S7A and B).

To evaluate the proliferation kinetics over time in the *IκBα* -deficient LT-HSCs, we studied the extent of BrdU label retention of HSCs from AGM to FL stage. After establishing that we could label around 80% of the LT-HSC regardless of genotype with two pluses of BrdU at E10 and E11 (Supplementary Fig. S7C and D), pregnant females with E10.5 embryos were pulse-labeled with BrdU and analyzed for label-retaining LT-HSCs in the FL of E14.5 embryos. We detected a higher frequency of BrdU+ cells in the *IκBα* KO LT-HSC compartment, corresponding to lower cycling events over time (Fig. 5D and Supplementary Fig. S8A). In short, all three approaches indicated a lower cycling rate of *IκBα* deficient LT-HSCs.

To determine whether the elevated RA signaling was driving the *IκBα* deficient LT-HSC low cycling status, we purified lin-depleted E14.5 FL cells from *IκBα* WT or KO embryos and treated them with either DMSO or 1 μM of the RARα inhibitor Ro-415253 (hereafter Ro-41) for 2 days. The cells were then examined for their cell cycle status (Fig. 5E). We found that treatment with Ro-41 increased the frequency of cycling cells both in the WT and *IκBα* KO LT-HSCs. Notably, treating *IκBα* KO LT-HSCs with a RARα specific inhibitor was sufficient to reduce the number of *G0* cells to similar levels of the *IκBα* WT (Fig. 5E and Supplementary Fig. S8B), further supporting the concept that increased RA signaling is the causal mediator of cellular dormancy in the *IκBα* LT-HSCs.

## *IκBα* KO LT-HSC shows a functional dormant phenotype in vivo and in vitro

Next, we aimed to investigate the activation capacity of LT-HSCs by measuring the clonogenic activity of individual *IκBα* WT and KO LT-HSCs in vitro (Supplementary Fig. S9A). LT-HSCs that are in a dormant/slow-cycling state need more time to proliferate than their already activated counterparts[18]. In fact, dormant HSCs show a paradoxical behavior upon stress stimuli, including transplantation assays or ex vivo culture: the cells become activated with a delay which has been attributed to their need to exit the quiescent state first. However, once

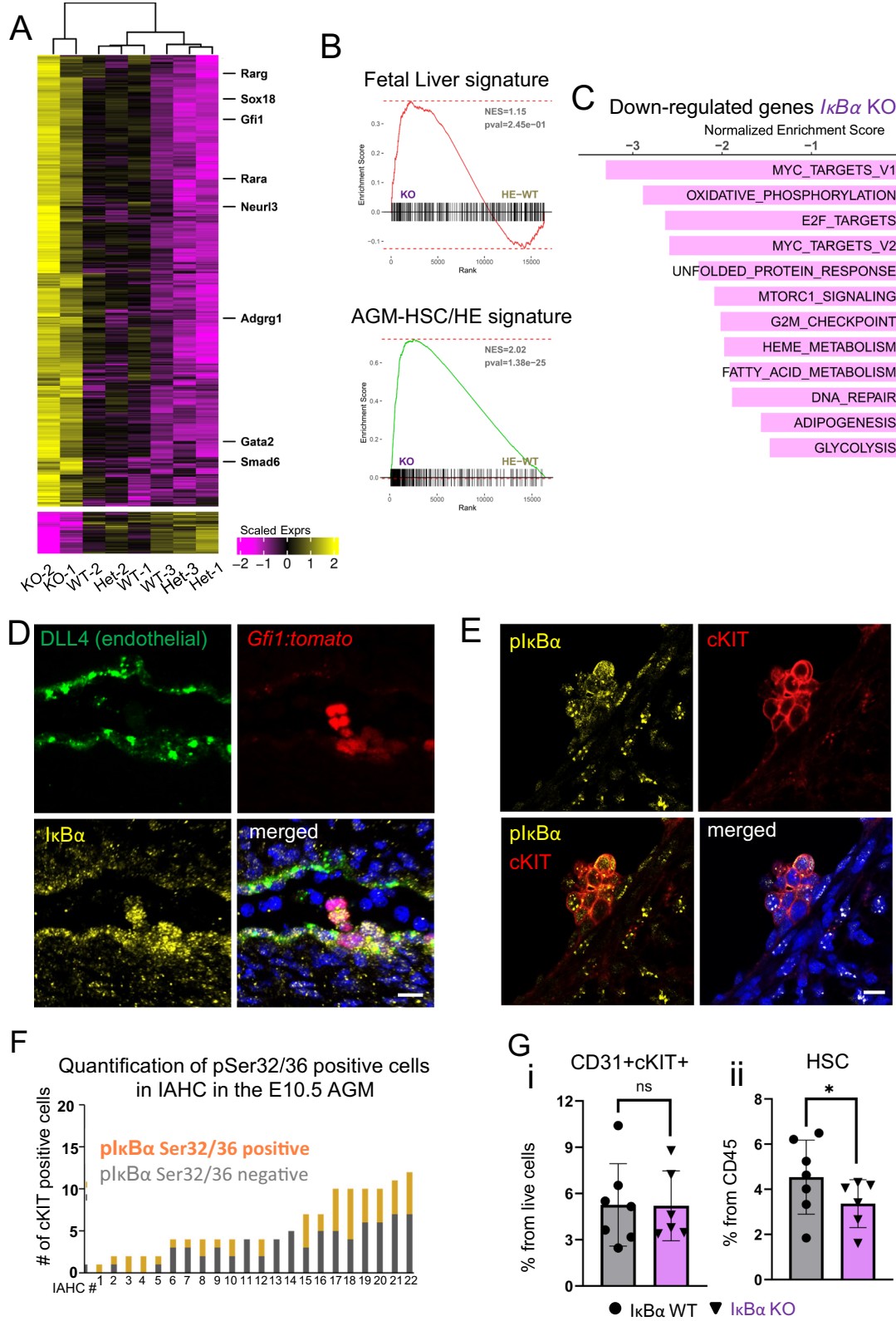

activated, dormant cells possess a higher self-renewal and repopulation capacity than their already activated HSC counterparts[49], and even outperform their WT equivalent over time[18,19]. We, therefore, compared the clonogenic potential of *IκBα* WT and KO LT-HSC in vitro by sorting single LT-HSCs into individual wells and assessing for the cell number of the clonogenic colonies after 10 days. We detected an overall trend towards increased colony numbers from *IκBα* KO LT-HSC

with the small colonies (10–100 cells) showing the highest increase (Supplementary Fig. S9Aii, iii), suggesting that *IκBα* KO LT-HSC have a delayed response to stress (more small colonies) but greater proliferation potential (higher overall number of colonies) (Supplementary Fig. S9A).

Ultimately, we assessed the dormant phenotype in serial transplantation settings (Fig. 6A). We evaluated the hematopoietic activity

**Fig. 3 | *IκBα* KO LT-HSCs maintain an AGM-specific HSC gene expression program. A** Heatmap of 1476 differentially expressed genes (DEGs) from the comparison of *IκBα* KO against WT/HET samples. DEGs were called with absolute shrunken logFC > 1 and adjusted *p*-values (Benjamini–Hochberg procedure, FDR < 0.05). A Wald-test was applied for DEG analysis. **B** GSEA comparing *IκBα* WT/HET and KO LT-HSCs bulk RNA-seq data against the top 300 genes defining (top panel) fetal liver LT-HSCs signature (Manesia et al.[42]) or (bottom panel) AGM HSCs (Thambyrajah et al. 2023[74]). Indicated *p*-value is a nominal *p*-value obtained from GSEA (**C**), Bar plot depicting significantly enriched MSigDB HALLMARK pathways in *IκBα* KO compared to WT/HET with negative Normalized Enrichment Score (NES) and identified by GSEA from the bulk RNA-seq data (Benjamini–Hochberg procedure (FDR) adjusted *p*-value < 0.05). **D** IHC on E11.5 (43–45 s) sagittal AGM section for endothelial marker (DLL4, green) and IκBα (yellow) and DAPI (blue) on *Gfi1:tomato* (red) embryos. Representative image from *n* = 4 WT and 5 KO AGM 12 um sections derived from 2 embryos for each genotype. Scale bar: 20 μm. Imaged using an SPE (Leica) with a 20× oil lens. **E** IHC on E11.5 (42–45 somites) cross-section of AGM for pSer32/36 IκBα (yellow), cKIT (red), and DAPI (blue). Scale bar: 10 μm. Images were taken using an SPE (Leica) with a 20× oil lens and processed using Imaris. **F** Bar chart depicting the number of p-IκBα Ser32/36 positive cells within cKIT+ IAHC at E10.5 (12 embryos of 33–36 somites). Each bar represents one IAHC. Counts were performed manually. Imaged using an SPE (Leica) with a 20x oil lens. **G** Bar chart depicting individual values for the frequency of (**i**) cKIT+(IAHC) in CD31+ cells and (**ii**) SCA1+EPCR+ positive cells within IAHC of E11.5 (42–46 somites) AGMs of *IκBα* WT (*n* = 7) and KO (*n* = 6) in 3 independent experiments. Statistical test: Multiple linear regression model with Experiment and Genotype covariates, significance obtained from applying t-test to corresponding model coefficient estimate (*\*p*-value < 0.05, ns *p*-value > 0.05). Bars indicate mean values and error bars refer to ± standard deviation. Source data are provided as a Source Data file.

of *IκBα* KO LT-HSCs (200 per recipient) or LSK (1000 cells) in competition with 500.000 lin- depleted bone marrow WT cells, respectively, in transplantation assay (Fig. 6A). Indeed, PB blood analysis 4 weeks after the transplantation showed a significantly lower blood chimerism from *IκBα* KO LT-HSCs (Fig. 6B) and LSK cells (Supplementary Fig. S9 B, C). But in the following bleedings (8, 12, and 16 weeks), the *IκBα* KO cells progressively reached a similar level of engraftment to their WT counterparts (Fig. 6B). By 4 months after transplantation, there was no significant difference between the genotypes in their ability to contribute to lineage differentiated cells in recipients (Supplementary Fig. S9E). We further conducted secondary transplantations (Fig. 6A and Supplementary Fig. S9B) and detected a trend to a higher repopulating capacity of *IκBα* KO over time that reached statistical significance by 3 months after the transplantation (Fig. 6C and Supplementary Fig. S9D). Moreover, all blood lineages were reconstituted by both *IκBα* WT and KO LT-HSCs in primary and secondary recipients (Supplementary Fig. S9E). We noted enhanced donor contribution to the LT-HSC population of secondary recipients from *IκBα* KO (primary) LT-HSC donors (Fig. 6D). Thus, we further assessed the self-renewal capacity of *IκBα* KO LT-HSCs in tertiary transplantations. Remarkably, 30% of the *IκBα* KO LT-HSC continued to contribute significantly (>10% chimerism) to the peripheral blood of the tertiary recipients after 8 weeks, whereas none of *IκBα* WT LT-HSC recipients showed donor chimerism after 3 serial transplantations (Fig. 6E and Supplementary Fig. S9F), supporting the fact that more *IκBα* KO LT-HSC are in a deep quiescent state.

In summary, we have shown that loss of IκBα deregulates RA signaling in a PRC2-dependent manner in the embryonic HSC compartment (please see model in Fig. 6F), from the onset of their specification in the AGM to the fetal liver stages. Consequently, we detect less LT-HSCs, and the remaining LT-HSCs in *IκBα* KO exhibit molecular and functional characteristics of dormant HSCs.

## Discussion

NF-κB is a pleiotropic transcription factor that responds to inflammation by activating multiple biological processes in emergency situations. Inflammatory signals in turn are coordinated with cellular differentiation, however the mechanisms underlying this crosstalk are not well understood. On the contrary, the environment in the embryo is generally protected from insults. Nevertheless, inflammatory signals, including canonical NF-κB signaling are critical for HSPC formation in the AGM[30,50], implying that there must be safeguards to protect LT-HSCs from extensive differentiation, whilst more committed progenitors undergo differentiation and proliferation.

Although the main function of IκBα is the inactivation of NF-κB complex by cytoplasmic retention or by active removal from the nucleus[51], there is an ancestral function of IκBα that is evolutionary conserved, it is in concert with the PRC2 complex and targets stem cell-specific genes[32–34]. Our data now show that this nuclear IκBα-PRC2 activity could be a deterministic factor for LT-HSCs in response to inflammatory cytokines already in the AGM.

Here, we report that *IκBα* deficient embryos can specify all blood lineages and progenitors, but they show a significant decrease, particularly in the number of LT-HSCs generated at different developmental stages. Remarkably, we detect a decline in the most *naïve* HSC pool in the E11.5 AGM already, indicating a requirement for IκBα-PRC2 interactions at this early stage. The small number of *IκBα* KO LT-HSC that are present in the FL at E14.5 retain an embryonic/AGM-specific signature, including *Sox18*, *Gpr56*, and *Gata2*, suggesting an impaired maturation and/or low proliferation. However, we cannot exclude that HE specification may also be compromised.

These observations agree with earlier findings in the *IκBα* deficient adult intestinal stem cells that maintain a fetal signature[33]. In both scenarios, the adult stem cell programs are compromised in the absence of IκBα. Conditional deletions of *IκBα* at specific developmental times are still required to assess if the fetal stem cell population will be affected if the embryonic program is conserved.

Interestingly, we find that metabolic processes typical for adult low cycling/dormant cells, i.e., lower levels of Oxidative proliferation, and glycolysis are enhanced in *IκBα* KO FL HSCs while showing reduced expression of MYC/E2F targets. In further support for a low cycling state, we uncovered a molecular dormancy signature in *IκBα* KO LT-HSC that is akin to dormant bone marrow HSCs. Although HSCs are reported to enter the dormant state around 3 weeks after birth[52], we detect cells compatible with a dormant phenotype already in the AGM in the absence of *IκBα*. Our results thus suggest that a small population of HSCs may be acquiring the dormant phenotype already in the AGM that is regulated by RA, and this population is amplified in the absence of IκBα. These findings agree with the latest reports supporting the existence of a less proliferative HSC pool generated in the AGM[16,17].

In support of the molecular profiling, *IκBα* KO LT-HSCs cycle less as quantified by Ki67 staining and BrdU labeling experiments. Functionally, *IκBα* KO LT-HSCs can exit the dormant state in stress situations, i.e., transplantations or in vitro culture. However, time is needed to exit the dormancy program. In our transplantation assays, we detect this process as a delay of *IκBα* KO LSK and purified LT-HSCs to contribute to the peripheral blood of the recipients at WT levels. But in secondary and tertiary recipients, *IκBα* KO cells outperform significantly. It will be interesting to determine the contributors to this enhanced HSC activity.

Using C&T experiments for H3K27me3 we reveal RA signaling as the main pathway targeted by the IκBα-PRC2 interactions. Recent studies have established that all HSC activity is contained within RA metabolizing cells in vivo and in vitro[23,46,53,54]. Indeed, RARα induction through stimulation with the RARα specific agonist AM580, or its inactivation through RO-415253 has been conclusively linked to HSC emergence and activity in the AGM[23,24,55].

From studies in the LSK compartment of adult bone marrow, RARγ has been proposed as a regulator of stem cell renewal[25,26].

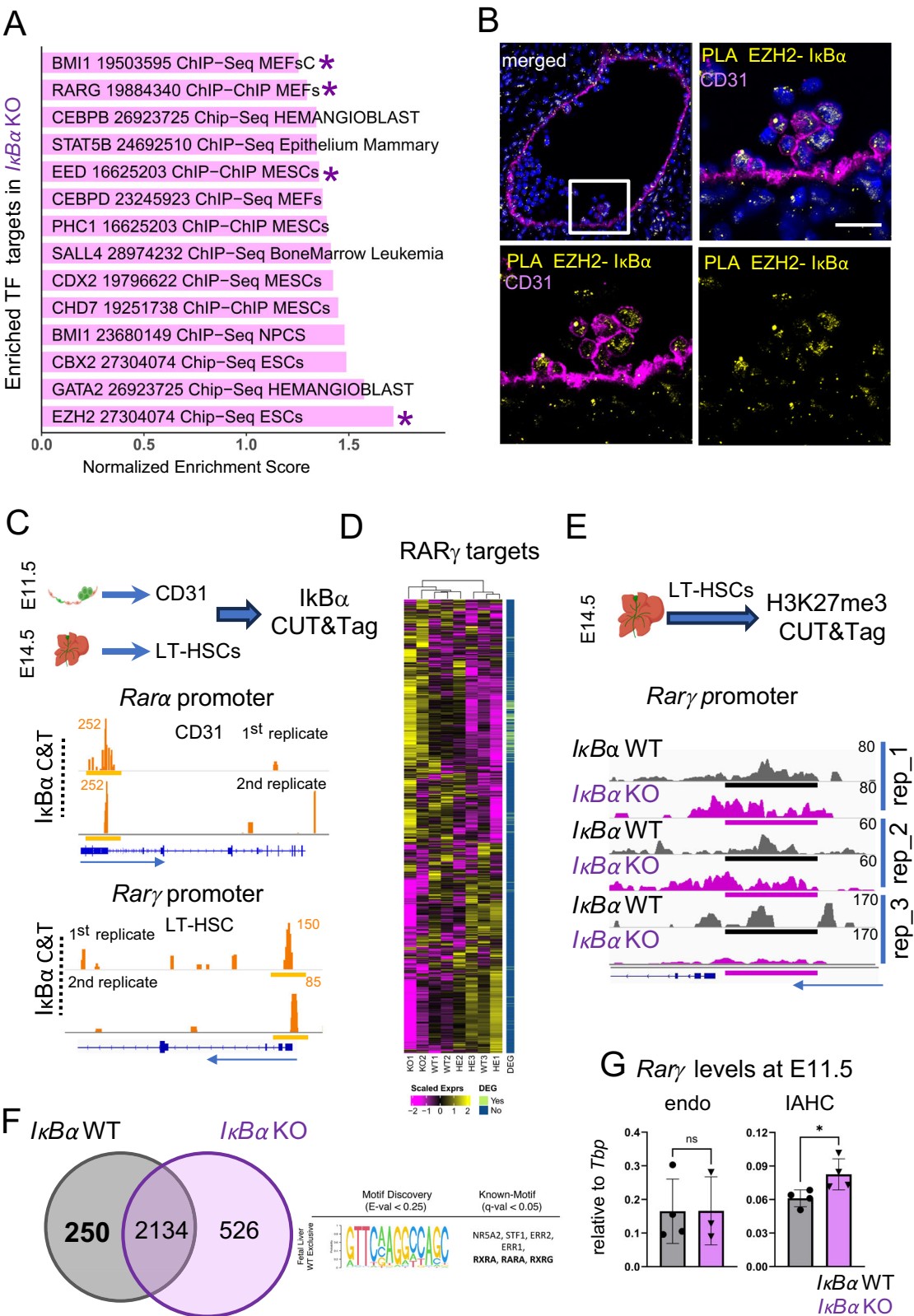

Moreover, RA levels have opposing effects on the stem cell compartment and more committed progenitors[56] and protect HSCs from excessive proliferation[26]. Recent data on purified adult LT-HSCs propose that the RA receptor RARβ is responsible for the switch between dormancy and activation[46]. In our RNA-sequencing data from FL LT-HSC, *Rarβ* levels are hardly detected. Based on all this data, we hypothesize that different isoforms prevail at distinct stages of hematopoiesis. Levels of RA signaling, maybe even through the different paralogs, could be responsible for fine-tuning the balance between quiescence and activation in LT-HSCs.

By controlling exactly this equilibrium, the IκBα-PRC2 axis can downregulate RA signaling precisely in HSCs to limit cycle entry when it is likely to be induced in response to inflammatory stimuli in the LT-HSC compartment. Conversely, in more committed progenitors where

**Fig. 4 | IκBα associates with the PRC2 complex and regulates retinoic acid receptors *Rarα* and *Rarγ*. A** Bar plot of enriched ChIP data sets from ChEA to in comparison to up-regulated genes in *IκBα* KO in the bulk RNA-seq data (FDR adjusted *p*-value < 0.05). NES: Normalized Enrichment Score, asterisks = TF mentioned in the text. **B** Proximity Ligation assay (PLA) for IκBα and EZH2 on E11.5 (43–46 somites) AGM sections (yellow), endothelial cells (magenta), and nuclear staining (DAPI). Scale bar: 20 µm. Representative image of 3 embryos in 3 independent experiments. Imaged with Stellaris8 (Leica) using a 20× oil lens. **C** Cut and Tag (C&T) for IκBα on E11.5 AGM (42–46 somites) derived CD31+cells (*n* = 35 embryos) and E14.5 LT-HSCs (*n* = 15 embryos). Read pile-ups in the IGV browser at the promoters of *Rarα* (CD31+) and *Rarγ* in LT-HSCs are indicated with a yellow line (*n* = 4 independent experiments). **D** Heatmap showing the expression levels of *RARγ* target genes (TRANSFAC TF:M10353) in the RNA-seq samples of E14.5 LT-HSCs from *IκBα* WT/HET and KO. vertical bar = identified as DEG (light green). **E** C&T for H3K27me3 in E14.5 LT-HSCs from *IκBα* WT and KO. Depicted are reads pile-ups in the IGV browser at the promoter of *Rarγ* (peaks regions, black line = WT, magenta line = KO). *n* = 5 WT and 8 KO embryos in two independent experiments. **F** Venn diagram of genes associated with the H3K27me3 peaks detected in E14.5 LT-HSCs of *IκBα* WT and KO. 457 and 854 genes are exclusively detected in *IκBα* WT and KO, respectively. Discovered motif (STREME) and associated TF are shown for the unique peaks in *IκBα* WT. **G** quantitative PCR (qPCR) for *Rarγ* expression levels comparing *IκBα* WT and KO derived E11.5 (42–45 somites) CD31+cKIT- (endothelial cells, WT *n* = 4, KO *n* = 3) and CD31+cKIT+ (IAHC, WT *n* = 4, KO *n* = 4) samples obtained from *n* = 2 AGMs per genotype in one experiment. Statistical test: unpaired one-tailed Mann–Whitney *U*-test (*\**p*-value < 0.05, ns *p*-value > 0.05). Bars indicate mean values and error bars refer to ± standard deviation. Source data are provided as a Source Data file. **C, E** created with BioRender.com released under a Creative Commons Attribution-NonCommercial-NoDerivs 4.0 International license.

we cannot detect this function, the response to inflammatory stimuli can be divergent and include a prolonged activation/proliferation.

Finally, we cannot exclude that classical NF-κB transcriptional activity is also involved in regulating the HSC quiescence or activation. It is noteworthy that NF-κB regulators may likely control chromatin IκBα, since inflammatory signals such as TNFα regulate chromatin-bound IκBα[34]. The *IκBα* KO pups die from day 7–10 after birth from a p65-dependent severe skin inflammatory reaction[57]. Considering our results, we believe that in the embryo, the inflammatory response due to NF-κB over-activation is not initiated; however, in conditions where cells are exposed to external stimuli, NF-κB response may become decisive for survival.

Altogether, we postulate that LT-HSCs must interpret the inflammatory signals that they receive in the AGM differently to HPC through the nuclear IκBα-PRC2 complex. But how this difference in nuclear IκBα activity is achieved remains to be understood.

We are yet to fully uncover the incorporation of several inflammatory signals into a cellular response. It will be intriguing to define the "inflammasome" that allows the entry and exit out of a dormant state from HSCs which would significantly improve our basis for regenerative medical approaches.

## Methods
### Mouse lines and animal work
Animals were kept under pathogen-free conditions, and all procedures were approved by the Animal Care Committee of the Parc de Recerca Biomedica de Barcelona, license number 9309 approved by the Generalitat de Catalunya. Mice born in the Barrier area are socially housed, up to four male and five female mice, in 1145T (Tecniplast) cages in individually ventilated cages. Autoclaved black poplar shavings (Souralit) are used as bedding, and irradiated tissues as nesting material. Autoclaved cardboard cylinders are added to cages with mating pairs and individually housed mice as additional environmental enrichment. Once a week mating pairs, and once every two weeks the rest, of the socially housed mice, together with the nesting material, are transferred to clean cages. Mice have ad libitum access to autoclaved water and irradiated diet (J. Rettenmaier & Sohne GmbH + CO KG; RM3 for breeding pairs and young mice until nine weeks old, and RMl as maintenance diet after nine weeks of age). Rooms are maintained under standard environmental conditions (humidity: 40–60%; temperature: 20–24) and a 12 h light/dark cycle (lights on at 08:00 h). Animal care and use program is approved by the PRBB-Ethics Committee and accredited by AAALAC International, following European (2010/63/UE) and Spanish (RD 53/2013) regulations. The *Gfi1:tomato*[11] and *IκBα* targeted knockout mouse line (*Nfkbia*[tm1Bal], Jackson Laboratories, stock #002850) were used in this study. For time mating's females between the ages of 2–6 months and males between the ages of 2–12 months were used and *Gfi1:tomato* and/or *IκBα* HET and females were mated to *Gfi1:tomato* and/or *IκBα* HET males. Vaginal plug detection was considered as day 0.5. The resulting embryos were genotyped and used for downstream assays. In transplantation assays, C57BL/6 (CD45.1) males and females between the ages of 8–12 weeks were subject to irradiation and were housed as described here.

### Fetal liver and bone marrow transplantation experiments
Charles River C57BL/6 (CD45.1) was used as recipients after two rounds of irradiation at 4 Gy (males) or 4.5 Gy (females) or 4.5 (8 Gy or 9 Gy in total, respectively). In each experiment, same-sex animals between the ages of 8–12 weeks were used. FACS isolated 1000 LSK cells or purified 200 LT-HSC (LSKCD48-CD150+) IκBα WT, and KO (CD45.2) fetal liver were retro-orbitally injected with 500.000 lineage-depleted bone marrow cells as a competitor (CD45.1) in primary transplantation assays. In secondary recipients (CD45.1), the bone marrow of the primary recipients (CD45.2) was extracted at 16 weeks and a total of 1 million cells were transplanted into each secondary recipient after two doses of 4 Gy of irradiation. Peripheral blood (PB) donor chimerism was analyzed by FACS at 4, 8, 12, and 16 weeks. Lineage analysis on BM-derived cells was performed at 16 weeks post-transplant by flow cytometry with specific antibodies. Please see Supplementary Data 5 for a list of antibodies. The samples were analyzed on a Fortessa or LSRII instrument and FlowJo v10.

### Single-cell RNA-seq data analysis from Zhou et al.[13]
Single-cell RNA-seq data from Zhou et al.[13] (https://www.nature.com/articles/nature17997) was downloaded from Gene Expression Omnibus (GEO), accession number GSE67120. The downloaded SRR files were converted to Illumina paired-end fastq using the SRA Toolkit (version 3.0.1) using the function "fastq-dump –split-e" with default parameters. We subsequently mapped the fastq files with STAR (REF - https://pubmed.ncbi.nlm.nih.gov/23104886/) (version 2.7.9a) using the STARsolo to the mouse reference genome on 10x genomics web-page (https://cf.10xgenomics.com/supp/cell-exp/refdata-gex-mm10-2020-A.tar.gz). The STARsolo mapping and quantification command were with the following parameters; soloUMIdedup Exact, soloStrand Unstranded, soloFeatures Gene GeneFull, soloMultiMappers EM, out-SAMtype BAM SortedByCoordinate. Following mapping, the output of STARsolo was loaded into R (version 4.1.0) using the DropletUtils package (version 1.12.1). We obtained 262 cells from the public repository and retained cells with <15% mitochondria reads, leaving 261 cells. Cell identify was taken directly from the Zhou et al.[13] metadata with the following population numbers.

AGM E11.0 Endothelial (*n* = 16); AGM E11.0 T1-preHSC CD201 negative (*n* = 42), AGM E11.0 T1-preHSC CD 201 high (*n* = 28); AGM E11.0 T2-preHSC CD201high (*n* = 44); AGM E11.0 T2-preHSC (*n* = 32); Fetal liver E12.0 HSC (*n* = 21), Fetal liver E14.0 HSC (*n* = 32), Bone marrow adult HSC (*n* = 46).

The list of gene signatures (Nfkb members, inflammation) and scripts to process Zhou et al.[13] data can be found at: https://github.

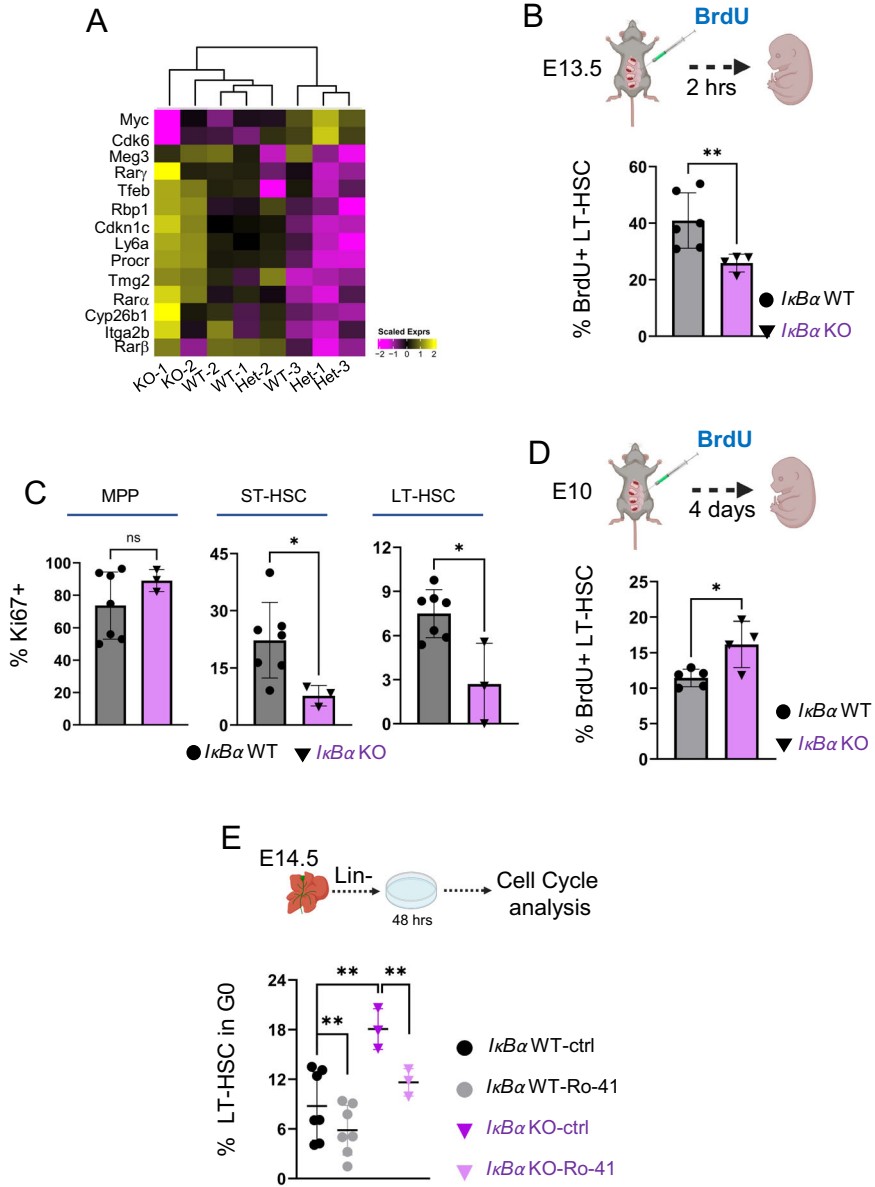

**Fig. 5 | *IκBα* KO (LT)- HSC are slow-cycling/dormant and can be mobilized by RARα inhibition. A** Heatmap showing the scaled expression levels of dormancy-associated genes (Cabeza-Wallscheid, 2017) from *IκBα* WT/Het and KO E14.5 LT-HSCs RNA-seq samples. **B** Bar chart with the frequencies of BrdU+ E14.5 LT-HSC after 2 h. in vivo labeling with BrdU in E13.5 *IκBα* WT (*n* = 6) or KO (*n* = 4) obtained from *n* = 10 embryos in 2 independent experiments. Statistical test: unpaired one-tailed t-test with Welch correction (**p*-value < 0.01). Bars indicate mean values and error bars refer to ± standard deviation. **C** Bar chart with percent of Ki67+ cells in indicated cell populations of E14.5 IκBα WT (*n* = 7) or KO (*n* = 3). MPP = multipotent progenitors, ST = Short-term HSCs. *n* = 7 embryos. Statistical test: unpaired one-tailed Mann–Whitney *U*-test (**p*-value < 0.05, ns *p*-value > 0.05). Bars indicate mean values, and error bars refer to ± standard deviation. **D** Bar chart with the percent of BrdU+ cells within the E14.5 FL LT-HSC after in vivo labeling of E10.5 *IκBα* WT (*n* = 5)

or KO (*n* = 4) embryos. Statistical test: unpaired one-tailed *t*-test with Welch correction (**p*-value < 0.05). Bars indicate mean values, and error bars refer to ± standard deviation. **E** Bar chart with the percentage of LT-HSC in *G0* phase of the cell cycle after 48 h of ex vivo culture of E14.5 *IκBα* WT (*n* = 7) or KO (*n* = 3) FL lin-cells treated with 10 uM of Ro-41 or DMSO. Data originates from 2 independent experiments with 10 embryos. Statistical test: paired one-tailed *t*-test for Ro-41 against DMSO comparisons and unpaired one-tailed *t*-test with Welch correction for WT against KO, in DMSO, comparison (**p*-value < 0.01). The horizontal lines indicate mean values and error bars refer to ± standard deviation. Source data are provided as a Source Data file. **B**, **D**, **E** were created with BioRender.com released under a Creative Commons Attribution-NonCommercial-NoDerivs 4.0 International license.

 Two group comparisons were performed using the *t*-test comparison using the ggsignif package (version 0.6.4).

### Genotyping PCR
Small pieces of embryonic tissue or yolk sac were dissected off the embryo and placed in PCR tube containing 30 µl of PBS. The tissue pieces were boiled for 8 min at 98 °C for denaturation. The tissues were digested with Proteinase K (50 µg/ml) for 30 min at 50 °C,

and the enzyme deactivated by boiling the samples for a further 10 min at 95 °C. 1 µl of the samples was used as a template for the PCR.

### AGM dissection, single-cell suspension
AGMs of E10–E11.5 embryos were dissected in PBS with 7% fetal calf serum (FBS) and penicillin/streptomycin (100 U/mL). Single-cell suspensions were generated by incubating the tissues for 20–30 min in 500 µl of 1 mg/ml of Collagenase/Dispase (Roche cat# 10269638001)

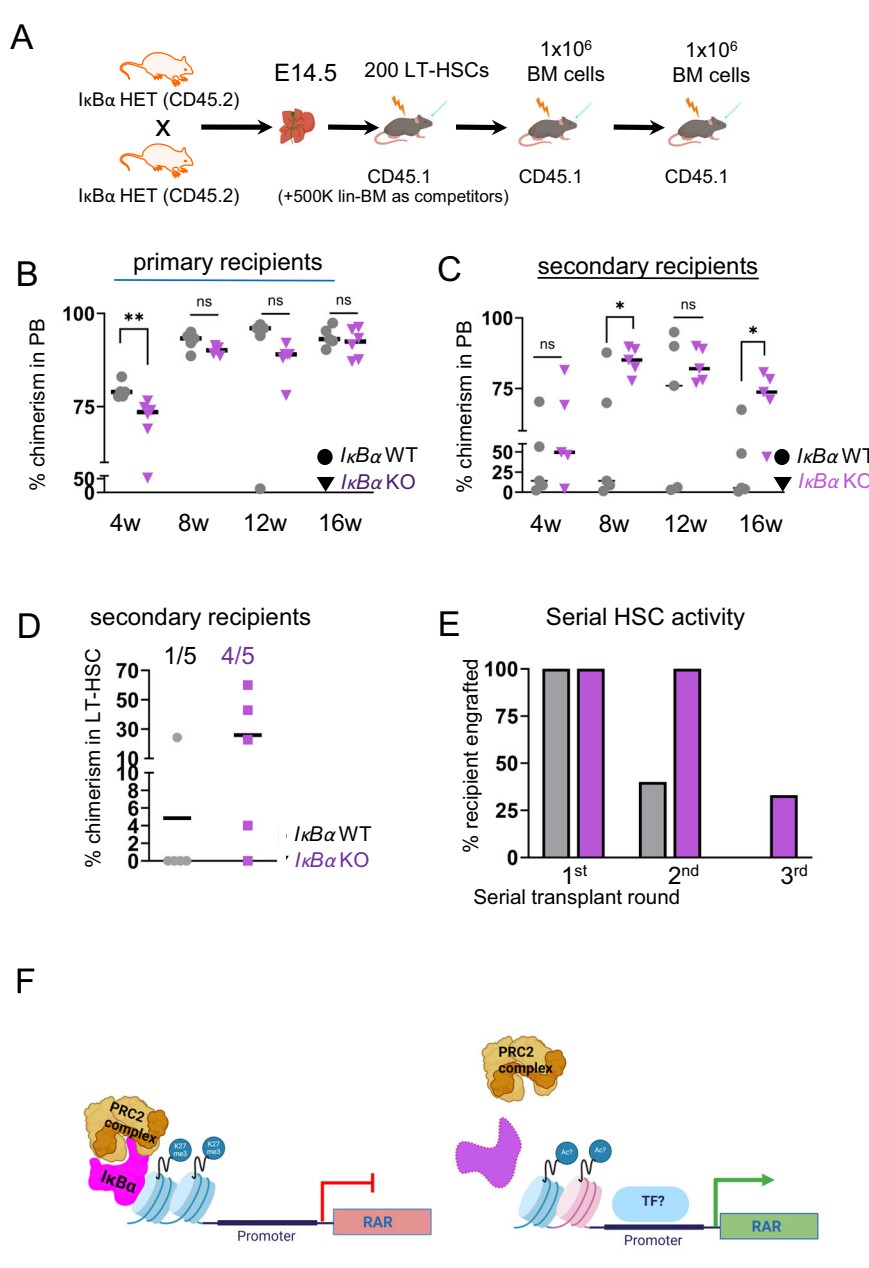

**Fig. 6 | IκBα KO LT- HSC are functionally slow cycling/dormant and are mobilized in stress situations. A** Scheme of the transplantation experiment. LT-HSCs were FACS purified from E14.5 *IκBα* WT and KO fetal liver (CD45.2) and transplanted serially into lethally irradiated donors (CD45.1). **B** Scatter plot with the percentage of donor chimerism (CD45.2) in the peripheral blood of primary recipient (CD45.1) in 4-week intervals with *n* = 5 WT and *n* = 6 KO recipients. Recipients of 200 purified LT-HSCs from E14.5 fetal liver of pools of 3 WT and 4 KO *IκBα* embryos. Statistical test: unpaired one-tailed Mann−Whitney *U*-test for all pairwise comparisons (**p-value < 0.01, ns *p*-value > 0.05). The horizontal lines indicate mean values. **C** Scatter plot with the percentage of donor chimerism (CD45.2) in the peripheral blood of the secondary recipient (CD45.1) in 4-week intervals. Each secondary recipient received 1 × 10⁶ nucleated bone marrow cells from individual primary recipients with *n* = 5 WT and *n* = 5 KO recipients. Statistical test: unpaired one-tailed Mann−Whitney *U*-test for all pairwise comparisons (**p*-value < 0.05, ns *p*-value > 0.05). The horizontal lines indicate mean values. **D** Scatter plot showing the percentages of donor chimerism (CD45.2) in the bone marrow LT-HSC compartment of secondary recipients (from C) at 16w. The horizontal lines indicate mean values. Ratios indicate the number of animals with chimerism >1% **E** Bar chart depicting the percentage of animals reconstituted with more than 10% of donor cells (*IκBα* WT or KO) at each round of serial transplantation. **F** Schematic summary of the main findings **i** *IκBα* associates with PRC2 complex at RAR gene promoters and silences RA signaling after a stress stimulus. **ii** in the absence of *IκBα*, this silencing is lost, and RA signaling is hyper-activated. Source data are provided as a Source Data file. **A**, **F** created with BioRender.com released under a Creative Commons Attribution-NonCommercial-NoDerivs 4.0 International license.

before mechanical dissociation with a syringe and needle. The cell suspension was washed with IMDM + 10% FBS and for antibody staining (see Supplementary Data 5 for a list of antibodies). Samples were analyzed on a Fortessa or LSRII instrument (BD Biosciences) and further plotted using FlowJo V10.

**Lineage committed cell depletion from fetal liver and bone marrow**

Mouse bone marrow cells were obtained by extracting the femur and tibia bones and subsequently crushing the bones in cold PBS with 0.5% FBS Buffer using a mortar and pestle. Cells were washed with PBS + 5%

FBS and then labeled with a cocktail of antibodies against lineage markers (BD Bioscience, cat# 559971) before conjugation to anti-Biotin microBeads (Miltenyi Biotech, cat #130-090-485). Labeled cells were depleted using magnetic columns (Miltenyi Biotech), leaving behind isolated and unlabeled Lin− cells at once ready for downstream experiments. For FL, the tissue was harvested from newborn or E14.5 embryos and the dissected liver was crushed with the end of a 1 ml syringe through a 40 um cell strainer into IMDM + 10% FBS. For the FL and newborn stage, CD11b was not included in the lineage depletion mix.

## Flow cytometry and cell sorting
Sorts of bone marrow cells were performed on FACSAriaII or BD Influx (BD Biosciences) instrument after magnetic depletion of lineage-primed celled (Miltenyi Biotec) and staining with the appropriate antibodies in PBS with 10% FBS (please see Supplementary Data 5 for list of antibodies).

## Single-cell LT-HSC culture ex vivo
The fetal liver was harvested from E14.5 embryos and depleted from lineage-primed progenitors as described above. Subsequently, samples were stained for LSKCD48-CD150 and sorted as single cells (see above) into 96-well plates containing complete Stem Cell Medium (StemSpan SFM, LifeTechnologies containing 50 ng/ml IL-3, 50 ng/ml IL-6, 2 ng/ml FGF2, 50 ng/ml SCF, 25 ng/ml TPO, 30 ng/ml Flt3-Ligand (all Preprotech), 100 u/ml Penicillin/Streptomycin, 2 mM L-Glutamine). Cells were cultured in 96-well ultralow attachment plates for 10 days and colonies were scored under a light microscope.

## Ex vivo treatment of LT-HSC with Ro-41
The fetal liver was harvested from E14.5 embryos and depleted from lineage-primed progenitors as described above and were treated with either Ro-415253 (Sigma−Aldrich) (10 μM), or the respective amount of DMSO in 6-well plates for 2 days in the media described above.

## Cell cycle analysis
The FIX & PERM™ Cell Permeabilization Kit from Thermo Fisher (cat# GAS003) was used Ki67 staining. Samples were stained for LSKCD48-CD150 first and then fixed with Medium A for 15 min at room temperature. After a wash step, the intracellular anti-Ki-67 Alexa Fluor® 647 (Clone B56, BD Pharmingen) and DAPI were diluted in Medium B and stained for 30 min at room temperature. The samples were run on a Fortessa Instrument from BD Bioscience and analyzed with FlowJO V10.

## BrdU labeling/staining
2 mg of BrdU (in PBS) was injected in a total volume of 300ul into pregnant females. The fetal samples were processed and stained according to the manufacturer's recommendations (APC BrDU Kit, BD Bioscience, cat # 552598). First, FL was harvested and made into single-cell suspension as described above. The cells were lineage-depleted (as described above) and stained with the according cell surface antibodies, ie LSKCD48-CD150. Now, the cells were fixed for 15 min at room temperature (solution in the kit), followed by 2 washes and a DNaseI treatment for an hour at 37 degrees (in the dark) to digest the DNA expose BrDU. After a wash step, anti-BrDU and DAPI were added to the cells for staining (minimum of 30 min at 4 degrees). The samples were run on a Fortessa Instrument from BD Bioscience and analyzed with FlowJO V10.

## Immunohistochemistry
E10.5−E11.5 embryos were fixed in 2% Paraformaldehyde (Thermo Fisher) for 12 min before they were soaked in 30% sucrose overnight and mounted in an OCT compound. 12 μm sections were prepared using a cryostat. The sections were permeabilized in −20 degrees

Methanol for 12 min at −20 degrees, washed with PBS, and followed by serum blocking (PBS with 3% BSA, 0.5% FCS, 0.05% Tween20 and 0.25 mM MgCl₂) for 1 h before the sections were incubated with primary antibodies at 4 °C overnight in blocking buffer. Sections were washed three times in PBST (PBS with 0.05% TitonX100) for 15 min each and then incubated with fluorochrome-conjugated secondary antibody at room temperature for 1 h. Sections were further washed three times in PBST and mounted using Prolong Gold anti-fade medium with DAPI (Life Technologies). Images were taken using an SPE or Stellaris8 (Leica) with a 20× oil lens and processed using Imaris v4.8 or ImageJ v 1.54 h.

## Proximity Ligation assay (PLA)
E11.5 AGMs fixed in 2% Paraformaldehyde (Thermo Fisher) for 12 min, embedded in OCT, and cut into 12 um thick sections. The slides were permeabilized for 10 min in −20 °C 100% Methanol. Proximity Ligation assay was performed according to manufacturer's protocol (Sigma, cat# DUO96000). After washing with PBS the edges of the slides were Liquid blocker pen and several drops (~200 μL) of Duolink® Blocking Solution was applied to each slide and incubate in a heated humidity chamber for 60 min at 37 °C. The primary antibody was diluted 1: 200 in Duolink® Antibody Diluent and incubated in 150−200 ul/slide overnight at 4 degrees (antibody list Supplementary Data 5). Parafilm was carefully placed on top of the antibody mix to prevent evaporation. The following day, the slides were washed with the PLA wash Buffer A; and the secondary PLUS and MINUS PLA probes were applied in a total of 200 ul/slide, covered with Parafilm, and incubated in a preheated humidity chamber for 1 h at 37 °C. The slides were washed twice and a total of 200 ul/slide of ligation mix (from the kit) was added and incubate in a pre-heated humidity chamber for 30 min at 37 °C. To visualize the interactions, 200 ul/slide of amplification solution (from the kit) was added after 2 washes and incubated in a pre-heated humidity chamber for 100 min at 37 °C. Now, Buffer B was used for the final washes, and the slides were mounted in Fluoromount with DAPI for imaging. Images were taken on a Leica Stellaris with 20× oil lens and processed using Imaris v4.8.

## CUT&Tag assay
For CUT&Tag assays, either CD31+ (25.000-30.000 cells) from E11.5 AGMs or LT-HSCs from E14.5 fetal livers (3.000-4.000 cells of IκBα WT or KO) were FACS sorted and used for CUT &T ag assay of IκBα and H3K27me3, respectively. We followed the Benchtop CUT&TagV.3 (https://www.protocols.io/view/bench-top-cut-amp-tag-bcuhiwt6), an updated version from Kaya-Okur et al.[58] 2019 with their recommended antibody for H3K27me3 (for details, please see Supplementary Data 5) and loaded Tn5 from epicypher. In brief, cells were lightly fixed with 0.1% and incubate at room temperature for 2 min. The reaction was by addition of 1.25 M glycine to twice the molar concentration of formaldehyde and centrifuged for 4 min $1300 \times g$ at 4 °C and resuspend in Wash buffer (1 mL 1 M HEPES pH 7.5, 1.5 mL 5 M NaCl, 12.5 μL 2 M spermidine, bring the final volume to 50 mL with dH2O, and add 1 Roche Complete Protease Inhibitor EDTA-Free tablet) to a concentration of ~1 million cells per ml. Gently resuspend and withdraw enough of the ConA bead slurry such that there will be 10 μL for each final sample of up to 500,000 cells. Transfer ConA bead slurry into 1.6 mL Binding buffer (Mix 200 μL 1 M HEPES pH 7.9, 100 μL 1 M KCl, 10 μL 1 M CaCl₂, and 10 μL 1 M MnCl₂, and bring the final volume to 10 mL with dH₂O) in a 2 mL tube and mix by pipetting. Place the tube on a magnet stand to clear (30 s to 2 min) and repeat wash step again. Resuspend in Binding buffer (10 μL per sample) and hold the bead slurry at room temperature. Take the tubes with cells and add slowly to the bead slurry. Place on an end-over-end rotator for 10−15 min. Place the tubes on a magnet stand to clear and withdraw the liquid and resuspend cells in 50 μL of ice-cold Antibody buffer (Mix 8 μL 0.5 M EDTA and 6.7 μL 30% BSA with 2 mL Dig-wash buffer). Add 0.5 μL primary antibody to

each sample with gentle vertexing and nutate (overnight to several days at 4 °C). Liquid should remain in the bottom and on the side of the tube while rocking. To evaluate the success of the procedure without requiring library preparation, include in parallel a positive control antibody (e.g.α-H3K27me3), and optionally a negative control antibody (e.g., rabbit α-mouse IgG). Remove the primary antibody and mix the secondary antibody 1:100 in Dig-wash buffer (Mix 400 μL 5% digitonin with 40 mL Wash buffer) and squirt in 100 μL per sample while gently vertexing to allow the solution to dislodge the beads from the sides and nuate at room temperature for 60 min. After a quick spin, place the tubes on a magnet stand to clear and withdraw the liquid. Add 1 mL Dig-wash buffer. Invert 10× or gently vortex to allow the solution to dislodge most or all of the beads. Repeat wash twice and add pA-Tn5 adapter complex in Dig-300 buffer to a final concentration of 1:200 for 100 μL per sample and nutate at room temperature for 1 h. Clear and pull off the liquid and was three times with 1 mL Dig-300 buffer. Invert 10× or gently vortex to allow the solution to dislodge most or all of the beads. Now, add 300 μL Tagmentation buffer (Mix 5 mL Dig-300 buffer and 50 μL 1 M MgCl$_2$) and incubate at 37 °C for 1 h in a water bath or incubator. To stop tagmentation and solubilize DNA fragments, add 10 μL 0.5 M EDTA, 3 μL 10% SDS, and 2.5 μL 20 mg/mL Proteinase K to each sample and mix by full-speed vortexing ~2 s, and incubate 1 h 55 °C to digest (and reverse cross-links). The DNA can be cleaned using a Quiagen Spin mini-column. The clean DNA fragments (50 ul end volume) are used to directly generate sequencing libraries by using 21 μL DNA + 2 μL of 10 μM Universal or barcoded i5 primer + 2 μL of 10 μM uniquely barcoded i7 primer. In a thin-wall 0.5 ml PCR tube, using a different barcode for each sample*. Add 25 μL NEBNext HiFi 2× PCR Master mix. Mix, quick spin, and place in Thermocycler and begin cycling program with heated lid. 72 °C for 5 min (gap filling): 98 °C for 30, Cycle 3: 98 °C for 10 s Cycle 4: 63 °C for 75 s (combined annealing/extension step) for 13 cycles. Libraries were cleaned with SPRI beads and the samples were quantified and validated using a bioanalyzer and subsequently paired-end sequenced.

## CUT&Tag data analysis

Quality control was performed on raw data with FASTQC tool (v0.11.9). Raw reads were trimmed to remove adapters presence with TrimGalore (v0.6.6)[59]. Default parameters were used except for the required minimum quality and the stringency parameters which were set to 15 quality score and 3 bases, respectively. Trimmed reads were aligned to the GRCm38 reference genome with Bowtie2 aligner tool (v2.4.4)[60]. Reference genome was retrieved from Ensembl FTP site (http://ftp.ensembl.org/pub/release-102/). Bowtie2 was executed with the following parameters: --no-mixed –no-discordant –dovetail –very-sensitive-local. Multi-mapped reads and those reads showing MAPQ < 20 were filtered out. Peak calling was only conducted for the CUT &Tag H3K27me3 samples. For this purpose, duplicates were removed from obtained BAM files with Samtools (v1.15)[61]. Peak calling was performed with SEACR (v1.3) in a stringent mode without considering IgG control sample[62]. For this purpose, required input bedgraph files were obtained from deduplicated BAM files using bedtools (v2.30) following SEACR authors guidelines[63]. The top 0.5% of peaks were retrieved for downstream analysis. BigWig files were obtained from BAM files with duplicates and using the bamCoverage function from deepTools (v3.5.1) and the following parameters: --binSize 5 –smoothLength 60 –normalizeUsing RPGC –extendReads and --effectiveGenomeSize set to 2.31e9 (50 bp reads) or 2.41e9 (75 bp reads)[64]. Neither smoothing nor extension was used for obtaining bigWig files from CUT &Tag Ikba samples. Genes were annotated to identified enriched regions with the ChIPseeker R package (v1.10.3) considering a TSS region from −5 kb to +100 b[65]. A consensus peakset was obtained from the two fetal liver biological replicates per condition. For this purpose, the findOverlapsOfPeaks function from the ChIPpeakAnno R package (v3.32.0) was used with default parameters. Motif analysis was conducted with

MEMESuite CLI (v5.5.2)[66] with default parameters against HOCOMOCO v11 core mouse database[67]. Discovered motifs were retrieved from STREME tool[66]. For this purpose, called regions from SEACR in BED format were converted to FASTA sequences in chunks of 500 bp. Peak visualization was done with Integrative Genomics Viewer (IGV).

## Bulk RNA-seq data sequencing

In all experiments, we extracted total RNA from three mice per genotype using the mRNA ultralow input protocol for library preparation (Quiagen, cat# 180492). The RNA concentration and integrity were determined using Agilent Bioanalyzer [Agilent Technologies]. Libraries were prepared using standard protocols, and cDNA was sequenced using Illumina® HiSeq platform (50-bp single-end reads). After sequencing, one of the KO samples had to be discarded due to inadequate quality sequencing. For the rest of the samples (3xWT, 3xHET, and 2xKO), the sequencing depth ranged between 28 M and 36 M reads (average 32 M reads) per sample.

## RNA-seq data analysis

Quality control was performed on raw data with FASTQC tool (v0.11.9). Raw reads were trimmed to remove adapters presence with Trimgalore (v0.6.6)[68]. Reads were filtered by quality (Q > 30) and length (>20 bp). Trimmed reads were aligned to the reference genome with the STAR aligner tool (v2.7.8). STAR was executed with default parameters except for the number of allowed mismatches which was set to 1. The required genome index was built with corresponding GRCm38 gtf and fasta files retrieved from Ensembl (http://ftp.ensembl.org/pub/release-102/). Obtained BAM files with uniquely mapped reads were considered for further analysis. Raw gene expression was directly quantified in STAR (--quantMode GeneCounts option). The obtained raw counts matrix was imported into R Statistical Software environment (v4.2.1) for downstream analysis. Raw expression matrix included 55,487 genes per 8 samples in total. Experimental design considered three genotypes: IkBα KO (x2 female samples), IkBα WT (2× male and 1× female samples), and IkBα HET (1× male and 2× female samples). Prior to statistical analysis, those genes with less than 10 raw counts across the 8 samples under test were removed. After pre-filtering, 20,610 genes were available for testing. For visualization purposes, counts were normalized by the variance-stabilizing transformation method as implemented in DESeq2 R package[69] (v1.38.3). Additionally, the normalized expression matrix was gender-corrected with the removeBatchEffect function from the limma R package (v3.54.2)[70]. Differential expression analysis (DEA) was conducted with DESeq2. Fitted statistical model included sample gender and genotype. WT and HET samples were pulled together under the same condition and considered as the reference for the comparison test. Obtained log2 fold change values were shrunken with apeglm shrinkage estimator R package (v1.20.0)[71]. Raw p-values were adjusted for multiple testing using the Benjamini–Hochberg False Discovery Rate (FDR)[72]. Differentially Expressed Genes (DEGs) between KO and HET-WT samples were called with adjusted p-values (FDR) < 0.05 and absolute shrunken log2 Fold change > 1. Data visualization was performed with the ggplot2 (v3.4.1), complexHeatmap (v2.14.0), and EnhancedVolcano (v1.16.0) R packages.

## Functional analysis

Gene Set Enrichment Analysis (GSEA) was applied over the complete list of genes under test for DEA (20,610 genes). Genes were pre-ranked based on their shrunken log2 fold change. GSEA was conducted through fgseaMultilevel function from fgsea R package (v1.24.0)[73] with default parameters. Enrichment plots were generated with the same package. For this purpose, MSigDB Hallmark and ChIP Enrichment Analysis (ChEA) databases for *Mus Musculus* were interrogated. ChEA database was downloaded from https://maayanlab.cloud/Enrichr/#libraries (ChEA_2022). Corresponding

gene Entrez identifiers were used for the analysis. Benjamini−Hochberg procedure was used to obtain adjusted *p*-values. Enriched processes were called with adjusted p-values (FDR) < 0.05. Additionally, two specific signatures associated to fetal liver (FL) LT-HSC and AGM HSC-HE was also explored for GSEA (Manesia et al.[42]) (Thambyrajah et al.[41]). Briefly, FL LT-HSC signature was obtained from the conducted DEA between FL (E14.5) and adult bone marrow HSCs. We considered the top 300 DEGs (adjusted pval < 0.05 and abs (log2 fold change) >1) up-regulated in FL (out of 703 genes) and sorted per log2 fold change values. On the other hand, AGM HSC-HE signature was a combination of the top 300 representative genes from both HSC-HE and HSC clusters identified in AGM scRNAseq data. We considered those genes represented in at least 50% of the cluster cells with adj pval < 1e-05 and positive log2 fold change. The top 300 genes were the ones with the highest log2 fold change values.

**E14.5 LSK single-cell RNA-sequencing and data analysis**

**Data pre-processing.** 10X sequencing reads were demultiplexed and aligned according to the Cell Ranger pipeline (version 6.0.0) under default parameters. Sequences were aligned against the mouse GRCm38 reference genome to obtain feature-barcode matrixes, separately for WT and KO conditions. KO and WT gene count matrixes were merged and analyzed using the Seurat package (version 4.1.1) [1] in R (version 4.1.3).

**Quality control.** Cells were filtered based on the number of genes, removing those cells below a threshold located between the minimum and 1st quartile, according to the distribution of the number of genes per cell. Cells with more than 20% of mitochondrial gene content and genes not present in at least 3 cells were filtered.

**Clustering.** Cell-to-cell variability was normalized by the expression measurements using a scale factor of 10e4 followed by a log-transformation. Gene expression values were scaled and centered. The selection of highly variable genes (HVGs) was performed by evaluating the relationship of log(variance) and log(mean) and selecting the genes that exhibit the largest variance-to-mean ratio. We applied principal components analysis (PCA) to reduce the dataset dimensionality and we chose the significant principal components by assessing the ElbowPlot. The clusters were identified with FindNeighbors and FindClusters Seurat functions, selecting the Louvain algorithm. This step computes the k-nearest neighbors and creates the shared nearest neighbor (SNN) graph.

**Cell type annotation.** Cell types and states were annotated by exploring cluster-specific genes that were previously computed with FindAllMarkers function, using the Wilcoxon Rank Sum test and an average log2 fold change (log2FC) of 0.25. Each population was also explored for well-known markers genes to annotate the cell population (Supplementary Data 2). Those populations that were not possible to annotate were classified as technical clusters and removed from the dataset.

**Differential expression (DE) analysis WT vs. KO.** We performed differential expression analysis for each individual annotated population between WT and KO to find differentially expressed genes between both conditions, with a special focus on the naïve cluster. As the number of HSC cells is expected to be very few, we applied an average log2FC threshold of 0.15 and a minimum fraction of cells for a gene to be detected at 0.1.

**Trajectory analysis.** Monocle2 [2] was applied to order cells in pseudo-time based on the highly variable genes computed in C1/LT-HSC, C1/C2/ST-HSC (MPP1/MPP4), C2-C3_neu/MPP3 and C2-C3_Ery_MK/MPP2/4 populations.

**Statistical analysis.** Statistical analyses were performed using R software environment v4.3.1. Normal distribution was assessed using the Shapiro−Wilk test. The one-way ANOVA residuals were tested against normality distribution in the case of more than two groups under test. The statistical details are documented in the corresponding figure legend. Unless otherwise specified in the figure legend, an unpaired one-tailed student's *t*-test with Welch correction (parametric data) or Mann−Whitney *U*-test (non-parametric data) was applied. In case of 3 or more groups to test, a one-way ANOVA (parametric data) or Kruskal−Wallis (non-parametric) test was applied followed by a *t*-test or Dunn's test as post-hoc tests. Statistical significance was accepted when $p < 0.05$.

**Reporting summary**
Further information on research design is available in the Nature Portfolio Reporting Summary linked to this article.

## Data availability
Single-cell RNA-seq, bulk RNA-seq and CUT&Tag data generated in this study have been deposited in NCBI Gene Expression Omnibus (GEO) repository under GEO SuperSeries accession no. GSE188525, composed in respective SubSeries GSE214699 [https://www.ncbi.nlm.nih.gov/geo/query/acc.cgi?acc=GSE214699], GSE188523 [https://www.ncbi.nlm.nih.gov/geo/query/acc.cgi?acc=GSE188523] and GSE188524 [https://www.ncbi.nlm.nih.gov/geo/query/acc.cgi?acc=GSE188524]. Additionally, single-cell RNA-seq data from Zhou et al.[13] was downloaded from GEO with accession number GSE67120. Source data are provided with this paper.

## Code availability
Scripts that have been used to process data published in Zhou et al.[13] are deposited in Github repository: https://github.com/zakiF/PublishedPapers/tree/master/Nat_comms_Ikba Scripts that have been used to process the in house bulk RNA-seq and CUT &Tag assay are deposited in Github repository: https://github.com/BigaSpinosaLab/HSC_dormancy_Ikba_via_retinoic_acid.

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

## Acknowledgements

We thank all members of Espinosa and Bigas laboratories for helpful discussions and technical support. We also thank the animal facility, FACS facility, and genomic facility of the PRBB, CRUK Manchester Institute, and the single-cell Unit of the IJC for their technical support. This work was funded by grants from SAF2016-75613-R, PID2019-104695RB-I00, and PDC2021-120817-I00 from Agencia Estatal de Investigación (AEI) and SLT002/16/00299 from Department of Health, Generalitat de Catalunya. The work in G.L. laboratory is supported by Blood Cancer UK (19014) and Cancer Research UK Manchester Institute Core Grant (C5759/A27412). R.T. is a recipient of BP2016(00021) and BP/MSCA 2018(00034) fellowship programs from Generalitat de Catalunya/MSCA. M.M. is a recipient of a grant from the Instituto Carlos III, grant number CA22/00011 (co-funded by the European Social Fund Plus, ESF+, and by the European Union).

## Author contributions
A.B., R.T., and L.E. conceptualized the study, designed the experiments, and analyzed data. R.T., W.H.N., P.H., C.B., A.I., J.G., N.C., L.M. and C.R.-H. performed experiments and analyzed data. M.M. and M.P. analyzed the CUT&Tag data. M.C.-P. analyzed the L.S.K. scRNAseq under supervision of E.M. and M.E. Y.G., M.Z.F., G.L. analyzed the E11.5 A.G.M. scRNAseq and Y.G. and M.M. analyzed the bulk RNA-sequencing data of L.T.-HSCs. A.B., R.T., M.M., and G.L. wrote the manuscript.

## Competing interests
The authors declare no competing interests.
