## [Peer Review File · Nature Communications]

I κ B α controls dormancy in Hematopoietic Stem Cells via retinoic acid during embryonic developmentReviewers' Comments:

Reviewer #1:

Remarks to the Author:

In this study, Thambyrajah and colleagues report an NFkB-independent role of IκBα in the development of hematopoietic stem cells (HSCs) controlling HSC dormancy via regulation of the retinoic acid pathway.

Authors combined the characterization of IκBα deficient murine embryos, transplantation of embryonic hematopoietic cells and single cell molecular transcriptional profiling of HSCs and cut and tag assays for IκBα and H3K27me3 to support a role for the IκBα – retinoic acid signalling in HSC dormancy.

This is in agreement to previous reports on the role of Inflammatory NF-kB signaling in HSC specification during embryogenesis (Espín-Palazon, et al. Cell. 2014; 159:1070–1085) but it adds an NFkB-independent role of IκBα in HSC development, which from my point of view constitutes an important finding.

Still, I consider that there are aspects that require close attention by the Authors including:

1) I find major issues in the analysis/interpretation of the data, particularly regarding the description of a population of a “highly quiescent population of HSCs” in IκBα deficient embryos. BrdU FACS plots do not seem to properly assign cells to BrdU positive and BrdU negative fractions (Suppl Fig. 6A). Gates seem arbitrary and would be useful to show an Isotype control here. Gating and interpretation of flow cytometry plots are very different from other publications (e.g. Fig 2A in Petruccioli et al., Plos One 2011 Vorinostat Induces Reactive Oxygen Species and DNA Damage in Acute Myeloid Leukemia Cells). All G0 would be G1 instead. Moreover, cells not incorporating BrdU after a 2hour pulse may well be in G1 rather than G0 (Fig 5a). Furthermore, authors showed that LT-HSCs acquired BrdU over time, so cells are just cycling more slowly as authors indicate at some point in the text. But they are not dormant (Fig 5B). Thus, describing cells as dormant seems not appropriate. Even though in Fig 4E some genes related to quiescence were studied, further functional analyses for a G0 state would be indispensable (e.g. Ki67 analysis, which are indicated in the methods but not shown in the text). This could affect their major conclusions that the IκBα - retinoic acid (RA) “regulatory axis controls the balance between HSC proliferation and dormancy”.

2) In abstract and introduction, it seems that authors will be performing a deep study on the differences among HSCs and Hematopoietic Progenitor Cells (HPCs) based on recent findings indicating their potential independent embryonic origin. But authors just reported that they are able to find differences in HSC frequencies and not in HSPC frequencies in the embryo without going into any further investigation on this per se. No specific molecular studies were performed on embryonic HPCs to investigate the relevance of the IκBα - retinoic acid (RA) regulatory axis in those cells. Thus, either these differences are investigated or I think that the focus of the article should be better rephrased and differences among HPCs and HSCs not referred to.

In this same context, Authors indicated that HSCs “are generated as a safeguard pool of cells from the first stages of development” which seems still under debate for the developmental stages and should be better discussed for the reader.

3) Importantly, authors do not address the contribution to the observed phenotypes from potential non-cell autonomous effects potentially arising in IκBα germline deficient embryos. Especially taking into account the role of the NF-kB signaling pathway in dorso-ventral patterning and the development of liver (reviewed in Espin-Palazon and Traver, Experimental Hematology 2016;44:519–527). For instance, p65-knockout mice die at 15–16 days of gestation due to liver degeneration, which could greatly affect HSC survival, development and expansion in the fetal liver (FL) from E12. Authors performed transplants from E14 FL (Figure 5). Thus, a deep analysis on potential non-cell autonomous

effects seems essential.

4) The type of transplant performed in Fig 5D is not clear.

-Did authors compete FL-KO LSK with FL WT LSKs? or with adult BM LSK? Otherwise, it does not make sense to get chimerism of 75% for both WT and KO. Actually, in SF6 it indicates a transplantation protocol different from Fig 5D. Thus, it is not clear and if 11.5 AGM cells were transplanted too?

-In Fig SF6 it indicates that B-cells were studied in the spleen? Does showed data referred to PB or spleen?

Importantly, 1000 FL-LSK constitutes a saturating dose for FL-LSKs. Thus, at this level of saturation it seems difficult to effectively evaluate a competitive advantage/disadvantage. Moreover, authors seemed initially focused on HSCs rather than HPCs, which are within the LSK compartment. Thus, a transplant of sorted HSCs rather than LSKs seems more appropriate specially as they detected differences in short-term engraftment that could be derived from HPCs rather than HSCs.

Other issues that require clarification include:

1) Gene nomenclature is frequently not adequate in the manuscript (text and figures): e.g. "rara", "cdk6", "myc", "tfef" referring to mouse genes.

2) Fig 2E: The frequency of LT-HSCs is higher than the frequency of LSKs. How is this possible?

3) Words as "irrefutably" don't seem very appropriate.

4) Missing p-values in Fig. 1B. The highest inflammatory pathways seem enriched in adult HSCs. Any comment on this in the discussion?

5) Fig 2 and SF 2. Have results been normalized/stratified by somite pair counts (s.p.) of the analyzed embryos? s.p. are not indicated.

6) Fig 2E. Number of replicates for WT is too small to draw conclusions. "N" by genotype are not shown in the legend. Looks like 1 or 2.

7) Fig 3C,4D. STDev requires at least three values. Not the case for KO. Mixing WT and HET does not seem completely appropriate.

Reviewer #3:

Remarks to the Author:

Review of Thambyrajah et al.

For Nature Communications

Thambyrajah et al. present data that they argue demonstrates the $I\kappa B\alpha$ functions in emerging hematopoietic stem cells (HSCs) to promote proliferation by restraining expression of a retinoic acid (RA)-directed, pro-dormancy genetic program. They first present a re-analysis of published (Zhou et al., 2016) single cell HSC gene expression profiling and find that genetic elements of the $NF\kappa B$ inflammatory signaling program are expressed throughout hematopoietic ontogeny. This analysis is presented as the rationale for focusing on the functional contribution $I\kappa B\alpha$, often viewed as a repressor of $NF\kappa B$, but here posited as a direct regulator of chromatin methylation. They examine spatial distribution of $I\kappa B\alpha$ protein in the aorta-gonado-mesonephros (AGM) of E11.5 embryos and by flow at E10.5 to show that $I\kappa B\alpha$ shows distribution through a subset of HSCs. They present HSC frequency from an *Ikba* knockout, which they describe as having decreased frequency of embryonic emerging

HSCs (CD31+cKIT+CD45+GFI1+EPCR+) and decreased frequency of long term (LT) HSCs (LSK, CD48-CD150+ a.k.a. LSK-SLAM) at later stages of embryogenesis into adulthood. They perform comparative gene expression analysis of WT and *Ikba* KO HSCs and determine a set of genes that are upregulated (365) or downregulated (204) in the *Ikba* KO. They also examine H3K27me3 and find differential methylation that relies on $I\kappa B\alpha$ (uniquely up in WT) or on an unexplained mechanism (uniquely down in *Ikba* KO). Within the putatively $I\kappa B\alpha$ -dependent methylation regions they identify DNA-binding consensus sequences including for RA-responsive factors. They present a limited analysis of $I\kappa B\alpha$ occupancy compared to methylation status to support their model that RA-responsive gene upregulation is due to methylation alterations in a developmentally regulated manner. They examine the expression of a subset of putative dormancy signature genes (Cabezas-Wallscheid et al., 2017; Schönberger et al., 2022) and detect alterations in some of them, which they present as proof that differences in RA-responsive gene expression are responsible for putatively increased dormancy. They perform a series of cell cycle, BrdU studies, in vitro colony assays, and primary and secondary transplants that they present as support for increased dormancy in the *Ikba* KO HSCs.

Overall, the studies presented are interesting and address important questions: what is the genetic basis of functional heterogeneity of HSCs, how are functional differences established and maintained during ontogeny, and what are the main similarities and differences between embryonic and adult regulation of HSC behavior? The main conclusion of interest is that $I\kappa B\alpha$ promotes a distinct behavior within a heterogeneous population of HSCs, upstream of an NF κ B/inflammatory signaling-independent upregulation of RA-responsive genes to drive “dormancy” and the proposition that distinct RA-responsive cassettes mediate or partially mediate this behavior over the course of ontogeny. Our main concerns revolve around 1) lack of robust establishment of phenotypes, 2) lack or robust establishment of putative mechanistic explanations, 3) overinterpretation of results and inadequate methods, repetitions, and parallel means to establish conclusions, 4) unclear writing and presentation of data, 5) inadequate description and documentation of methodology. We believe that with major revision and additional experiments, these findings could be suitable for Nature Cell Biology and of high interest to its readers, assuming new results do not substantially alter the overall conclusions.

MAJOR

1. The putative physical mechanism of $I\kappa B\alpha$ inactivation of genes via PRC2 methylation and independent of NF κ B inhibition is not convincingly demonstrated, nor is the primary role of RA-responsive genes in this alleged phenotype.

2. The manuscript needs to be significantly edited for clarity and figures—especially schematics—need to be reworked to improve understandability.

3. We do not think examination of hematopoiesis in the P5/6 liver is meaningful. Hematopoiesis has moved to the BM by then. If the authors believe this tissue and stage are meaningful, documentation is needed and a clearer exposition of relevance.

4. Is looking at frequency of hematopoietic precursors meaningful? We believe that establishment of absolute numbers of HSPCs at stages examined needs to inform the percentage frequencies.

5. Overall, the HSC defects in the absence of $I\kappa B\alpha$ shown in this manuscript are not convincing. For example, the authors show that there is a decrease in the percentage of AGM CD31+cKit+CD45+GFI1+EPCR+ HSCs, but the representative flow plot differences are predicated on the addition of GFI1 as a marker (Fig. S2B). If just EPCR is considered, it appears there may be a higher number CD31+cKit+CD45+EPCR+ HSPCs in the *Ikba* KO. GFI1 expression in these populations has not been functionally shown to further purify HSCs and the papers referenced in the manuscript only show EPCR as a functional marker for HSCs, not GFI1. It is conceivable that there are simply lower levels of GFI1 in $I\kappa B\alpha$ embryos that do not correlate with HSC fate or function. Furthermore, the number of WT LSK and LT-HSC samples is too low and significant differences are only detected between HET and KO samples. The authors assume that there are no differences between WT and HET

samples (which is based on a very low number of replicates) and therefore it is ok to compare HET and KO samples as if they were comparing WT and KO samples. Lack of het phenotype needs to be established. This assumption has also been used for bulk RNA sequencing analyses, in which WT and HET HSPCs were pulled together.

6. Fig. 2F putatively shows a “worsening” diminution of LT-HSCs over development from E11.5 to P6/7. No data is presented on the frequency of LSK-SLAM at E11.5 or P6/7, so not clear where these ratios came from. The legend says “taken from 2Eii,” but there are no E11.5 or P6/7 numbers in those charts.

7. The exclusion of NF κ B signaling is not convincingly demonstrated. Evidence put forth that NF κ B pathway/inflammatory signaling is unaffected includes 1) Fig. S2B, which claims no elevation in p65 signal in ECs of I κ B α KO, but a) it looks like p65 might be up and b) IHC is not really an adequate approach to address alterations in p65. 2) Lack of elevated expression of NF- κ B target genes in WT vs. KO sequencing of E14.5 FL (Fig. S4B), but 7/12 genes show “significance” and several of them appear to show overt significance and possibly even a het phenotype. A credible piece of evidence: GSEA analysis shows upregulation of non-NF- κ B pathways (S4D labeled “F”).

8. The assertion that maturation of HSCs from an AGM-like gene signature to a FL-like gene signature—especially predicated on elevated expression of Gpr56, Gata2, Neurl3, Sox18, and Smad6—needs better support. Experiments or references to indicate that these genes are normally downregulated during this transition and that the “preserved” gene expression is similar to normal in the AGM. The statement, “this HSC pool is already compromised at the site of their origin in the E11.5 AGM since we find that the low number of I κ B α KO LT-HSC that are present in the FL at E14.5 retain embryonic/AGM specific signature” is moreover an overstatement. The authors need to present evidence that the problem is not due to failure of maturation in the FL.

9. The reliance on an unpublished AGM HSC (and HE) gene signature for the comparison demonstrating that I κ B α KO cells are more AGM-like is not really acceptable. The GSEA comparison parameters need to be better described and validated. Is comparing different size gene sets really meaningful?

10. No direct evidence that H3K27me3 alterations are directly due to I κ B α /PRC2 defects in the KO HSCs. CUT & TAG assay shows that H3K27me3 alterations exist in the KO, but the alleged mechanism has not been shown in this situation. The occupancy of the rara and rarg loci by I κ B α and its putative connection to “decreased methylation” in nearby regions of unknown significance in the gene bodies is not persuasive (Fig. 4A-C, Fig. S5B-C). The diminution in methylation is not particularly severe and of unknown functional consequence. If there is indeed decreased methylation, the dependence on I κ B α occupancy is unclear.

11. Cell cycle connection to injected BrdU (Fig. 5A-C) needs citations and controls to establish the methodology, efficacy, and validated timing for the conclusions drawn. For example, the authors need to describe how they translated BrdU incorporation to number of cells in G0 and established that a 2-hour post injection labeling period (5A) is meaningful. For pulse-chase BrdU (5B), validation of quiescence/dormancy based on injection and BrdU label-retention following a 4-day chase needs to be demonstrated as meaningful.

12. Although BrdU labeling experiments show significantly less BrdU incorporation in KO LT-HSCs, suggesting a more dormant phenotype, the functional results, in particular secondary transplantations are not convincing. In addition, if HSC dormancy is affected in the context of I κ B α deletion, it should be studied in the newborn bone marrow.

MINOR

1. Zhou reference duplicated.

2. This sentence—"In its [nuclear I κ B α] absence, RA levels remain elevated and prevent LT-HSCs to be activated"—does not accurately reflect the authors' model. Don't they mean, expression of RA-responsive genes, and not RA per se?
3. What is the meaning of the increased LSK frequency in the P5/6 BM? It's glossed as "comparable."
4. Fig. S4D mislabeled as S4F.
5. References needed for support of genes listed as "key embryonic (AGM) HSC-associated genes" including Smad6, Sox18, Gpr56, Gata2, and Neurl3? With the exception of Gata2 and possibly Gpr56, these are not the most obvious set of AGM HSC-associated genes. p. 7 top paragraph. What about Runx1? CD41? Gfi1? Gata2 not capitalized.
6. Gene/protein nomenclature, capitalization, italicization, spelling, and standardization editing needed throughout the text and figures.
7. Gpr56 label typo Fig. 3C.
8. FL data set from Manesia et al., needs better description in the text: how old? etc.
9. Color key needed for S4E.
10. What is the authors' proposed mechanism explaining the presence of uniquely H3K27me3 peaks in the I κ B α KO HSCs?
11. The schematic for Fig. 4A is very difficult to understand.
12. In the Methods section for BrdU labelling/staining, what does the "pf" in "2mg pf BrdU" mean?
13. Example flow cytometry plots and unstained/FMO controls for the gating should be shown.
14. There is a significant increase between HET and KO LSKs in P5 and P6 BM that is not mentioned at all; instead, frequencies of FL, BM and newborn liver LSKs are described as comparable.
15. Figure 5A does not show lower uptake of BrdU by KO LT-HSCs as stated in the text.
16. Figure 2 A, B – the number of embryos imaged should be added to the figure legend.

Reviewer#1

(Remarks to the Author):

This agrees to previous reports on the role of Inflammatory NF- κ B signaling in HSC specification during embryogenesis (Espin-Palazon, et al. Cell. 2014; 159:1070–1085) but it adds an NF κ B-independent role of I κ B α in HSC development, which from my point of view constitutes an important finding.

Still, I consider that there are aspects that require close attention by the Authors including:

1) I find major issues in the analysis/interpretation of the data, particularly regarding the description of a population of a “highly quiescent population of HSCs” in I κ B α deficient embryos. BrdU FACS plots do not seem to properly assign cells to BrdU positive and BrdU negative fractions (Suppl Fig. 6A). Gates seem arbitrary and would be useful to show an Isotype control here. Gating and interpretation of flow cytometry plots are very different from other publications (e.g. Fig 2A in Petrucci et al., Plos One 2011 Vorinostat Induces Reactive Oxygen Species and DNA Damage in Acute Myeloid Leukemia Cells). All G0 would be G1 instead. Moreover, cells not incorporating BrdU after a 2hour pulse may well be in G1 rather than G0 (Fig 5a). Furthermore, authors showed that LT-HSCs acquired BrdU over time, so cells are just cycling more slowly as authors indicate at some point in the text. But they are not dormant (Fig 5B). Thus, describing cells as dormant seems not appropriate. Even though in Fig 4E some genes related to quiescence were studied, further functional analyses for a G0 state would be indispensable (e.g. Ki67 analysis, which are indicated in the methods but not shown in the text). This could affect their major conclusions that the I κ B α - retinoic acid (RA) “regulatory axis controls the balance between HSC proliferation and dormancy”.

We thank the reviewer for this insightful comment. We now provide FMO controls for our gating strategy for determining the LT-HSC (LSK CD48-CD150) (A), including the FMO and Isotype control for the BrdU gate (B). The non-BrdU labelled sample is used as a reference to establish positive BrdU labelled cells. The difference in intensity labelling with the indicated referenced figure is likely due to in vitro labelling of a cell line (NB4) instead of in vivo labelling of primary HSCs.

We agree with the reviewer that indeed we do not know cell cycle stage of the BrdU unlabeled population. For this reason, we have now represented the BrdU positive population during the 2-hour chase period in which cycling cells will incorporate BrdU. We find the majority of the BrdU positive cells in the S phase (B, exemplary FACS plot from the 2 hr chase experiment) We therefore now compared the BrdU labeled population in I κ B α WT and KO and find a significantly reduced BrdU incorporation in I κ B α KO LT-HSCs (p-value 0.01), (New figure 5B and below).

As the reviewer already noticed, we did perform Ki67 staining on E14.5 fetal livers. We have now added this data in support of the reduced labeling of LT-HSCs with BrdU in 2 hrs (Figure 5B).

Our results indicate a reduced proliferation index specifically in the LT-/ST HSC compartment of $I\kappa B\alpha$ KO E14.5 fetal liver (Figure 5C). Thus, in summary, The $I\kappa B\alpha$ KO LT-HSCs are cycling at a lower rate than their WT counterparts.

Altogether, the genomic and functional characterization of $I\kappa ba$ -deficient HSCs agrees with a very low cycling HSCs that are metabolically and functionally similar to dormant HSCs.

2) In abstract and introduction, it seems that authors will be performing a deep study on the differences among HSCs and Hematopoietic Progenitor Cells (HPCs) based on recent findings indicating their potential independent embryonic origin. But authors just reported that they are able to find differences in HSC frequencies and not in HSPC frequencies in the embryo without going into any further investigation on this per se. No specific molecular studies were performed on embryonic HPCs to investigate the relevance of the $I\kappa B\alpha$ - retinoic acid (RA) regulatory axis in those cells. Thus, either these differences are investigated or I think that the focus of the article should be better rephrased and differences among HPCs and HSCs not referred to.

We apologize for this misconception presented in the initial manuscript. We have now re-written the entire manuscript with the primary focus on LT-HSCs, and their precursors from the AGM. We would also like to mention that all the studies that describe AGM hematopoiesis did not have the tools to specifically study the emergence of the HSC precursors, ie SCA1-EPCR population. Only very recent studies indicate that these precursors consist of an HSC population. The in-depth relationship between this SCA1-EPCR and (LT-) HSCs is not really defined.

We have attempted to further investigate the differences of LT- ST- HSCs and MPPs by single cell RNAseq in the FL (New Suppl Figure S4 and Suppl table T2), but we have not been able to determine major specific differences in I κ B α -deficient subpopulations.

In this same context, Authors indicated that HSCs “are generated as a safeguard pool of cells from the first stages of development” which seems still under debate for the developmental stages and should be better discussed for the reader.

We have now refrained from using this statement and instead phrase our hypothesis based on recent studies in the text as follows:

“Recent studies indicate that (LT-) HSC and HPCs fate is already segregated in the AGM, and that the fetal liver—serves as a niche for their amplification as separate populations, with the (LT)-HSCs pool remaining smaller than the extensively proliferating HPC pool (Patel et al., 2022; Yokomizo et al., 2022; Ganuza et al, 2022). Signals that allow this segregation, and how these LT-HSCs protect their integrity would help to understand this process.”

and

“However, it is unknown how, and at which developmental stage(s) this dormant state is induced.”

3) Importantly, authors do not address the contribution to the observed phenotypes from potential non-cell autonomous effects potentially arising in I κ B α germline deficient embryos. Especially taking into account the role of the NF- κ B signaling pathway in dorso-ventral patterning and the development of liver (reviewed in Espin-Palazon and Traver, *Experimental Hematology* 2016;44:519–527). For instance, p65-knockout mice die at 15–16 days of gestation due to liver degeneration, which could greatly affect HSC survival, development and expansion in the fetal liver (FL) from E12. Authors performed transplants from E14 FL (Figure 5). Thus, a deep analysis on potential non-cell autonomous effects seems essential.

This is a fair point, and we cannot rule out a non-cell autonomous effect of I κ B α deficiency on HSCs. However, lack of I κ B α specifically affects HCS cycling that can be rescued by inhibiting RAR α in HSC (Figure 5E). Besides, our molecular data supports a cell-autonomous regulation of I κ B α on RA signaling in HSCs.

We also expected that lack of I κ B α would result in higher NF- κ B activity, however several of the published phenotypes of the I κ B α KO animals suggest that loss of I κ B α in the embryo does not oppose, or recapitulate NF- κ B defects:

- The I κ B α KO animals survive the fetal stage and start to only show a growth reduction from 3 days after birth, and death only occurs at day 6/7 days of age due to severe widespread dermatitis when they are exposed to inflammatory cues (Beg et al, 1995; Klement et al, 1996).
- Apart from an increased percentage of monocytes/macrophages in spleen cells, the gross organogenesis seems intact (Klement et al, 1996).
- In the initial study on the I κ B α KO phenotype, the authors assessed the levels of nuclear p65, c-Rel, p105, p50, p100 and I κ B β that remained steady in all three genotypes in embryonic fibroblasts (I κ B α WT, Het and KO) (Beg et al, 1995). However, in hematopoietic cells, an up-regulation of some, but not all NF- κ B targets was observed. In the same study, the authors generated I κ B α -/-p50-/- mice, where theoretically, over-activation of NF- κ B should be prevented. Even these animals should have the same defect, but the onset of phenotype was delayed by 2-3 weeks after birth (Beg et al, 1995).

- Both studies conducted detailed analysis of all internal organs and did not mention any abnormalities in the newborn liver (Beg et al, 1995; Klement et al, 1996).

Nevertheless, we specifically analysed the E16.5 fetal livers of I κ B α WT and KO in case there was some defect associated to p65 activity.

Fetal liver at E16.5

We observed no gross abnormalities in the morphology, cellularity, or the weight of I κ B α KO fetal livers compared to their WT litter mates.

Additionally, we performed IHC for total p65 and (active) p65-Ser536 on E11.5 AGM and E14.5 Fetal liver sections. In both sites of hematopoiesis, we did not detect any obvious increased staining in I κ B α KO (Suppl Figure S6A-C).

4) The type of transplant performed in Fig 5D is not clear. -Did authors compete FL-KO LSK with FL WT LSKs? or with adult BM LSK? Otherwise, it does not make sense to get chimerism of 75% for both WT and KO. In SF6 it indicates a transplantation protocol different from Fig 5D. Thus, it is not clear if 11.5 AGM cells were transplanted too?

We apologize for the confusion.

- 1:1 BM cells was used as support and as the reviewer spotted, the FL LSK will perform better in the primary transplantation (Ganuza et al, 2022).

- We did perform AGM transplants, but they have not been included in the manuscript because there were too few cases. However, they results were consistent with our current transplantation data, ie recipients of IκBα KO AGMs were showed higher reconstitution in secondary transplantations.

- In this revised manuscript, we have now added transplantation assays with purified LT-HSCs from the E14.5 FL, and these were also transplanted in competition with equal number of BM cells.

-In Fig SF6 it indicates that B-cells were studied in the spleen? Does showed data referred to PB or spleen? Check if it is PB or spleen???

We analysed both the spleen and PB for B220. The data in Figure Suppl Figure S8D is derived from the PB.

Importantly, 1000 FL-LSK constitutes a saturating dose for FL-LSKs. Thus, at this level of saturation it seems difficult to effectively evaluate a competitive advantage/disadvantage. Moreover, authors seemed initially focused on HSCs rather than HPCs, which are within the LSK compartment. Thus, a transplant of sorted HSCs rather than LSKs seems more appropriate especially as they detected differences in short-term engraftment that could be derived from HPCs rather than HSCs.

We have now performed transplantation experiments with purified LT-HSC (LSKCD48-CD150+) and, as with the transplantation assay performed with the LSK cells, we see a delay for the IκBα KO LT-HSC to catch up to WT levels in primary transplantations. When LT-HSCs are transplanted, the delay is shorter than when LSK cells are transplanted. We reason that in the LSK of the IκBα KO we are transplanting less LT-HSCs than when we equalize the numbers of LT-HSCs by transplanting purified LT-HSCs.

We have now moved the transplantation assays performed with LSK cells to the suppl Figures S8A-C and placed the transplantation assays with LT-HSCs in the main Figure 6A-E.

In the secondary transplantations, initially both genotypes are comparable, but IκBα KO recipients have a more sustained hematopoietic contribution at 16 weeks. Moreover, we find more donor derived chimerism in the bone marrow LT-HSC compartment of secondary

recipients. Finally, we also see more recipients engrafted in tertiary transplantations. In conclusion, all our transplantation assays point to a better reconstitution ability of I κ B α KO HSCs.

Other issues that require clarification include:

1) Gene nomenclature is frequently not adequate in the manuscript (text and figures): e.g. “rara”, “cdk6”, “myc”, “tfec” referring to mouse genes.

We have made these changes. Thank you for the suggestion.

2) Fig 2E: The frequency of LT-HSCs is higher than the frequency of LSKs. How is this possible?

We apologize for this inaccuracy. For the frequency determinations, we are referring to the previous gate, ie in the case of LSK, we are plotting the frequency derived from the lin- cells. In the case of LT-HSCs, we are showing the frequency derived from the LSK gate. Therefore, there is no direct correlation between the frequencies of LSK and LT-HSCs since we are not referring to the frequency from the total number of cells. We agree that we should have included this in the graphs, and we have now amended it accordingly.

3) Words as “irrefutably” don’t seem very appropriate.

We have now re-phrased sentences to be more objective.

4) Missing p-values in Fig. 1B.

We have added p-values.

The highest inflammatory pathways seem enriched in adult HSCs. Any comment on this in the discussion?

We have now directly described this observation in the results: “Similarly, the signatures of “inflammatory pathway”, “NF- κ B signaling activity” and NF- κ B signaling members themselves (Suppl table T1), were heterogenous during embryonic development, with a subtle decrease at fetal stages, but highly represented in the adult bone marrow HSC fate.”

5) Fig 2 and SF 2. Have results been normalized/stratified by somite pair counts (s.p.) of the analyzed embryos? s.p. are not indicated.

We have now included somite stages in the figure legends.

6) Fig 2E. The number of replicates for WT is too small to draw conclusions. “N” by genotype are not shown in the legend. Looks like 1 or 2.

The reviewer is correct. We have now added more data points for the WT genotype in now Figure 2B (and below) and demonstrate that there is no significant difference in Het compared to WT, but we still have a significant difference in the frequency of LT-HSCs between WT/Het compared to KO. Here, we would like to also point out that our RNA sequencing data sets comparing gene expression between I κ B α Het and WT LT-HSCs also indicated that there is not a haploinsufficiency phenotype for I κ B α Hets.

7) Fig 3C,4D. STDev requires at least three values. Not the case for KO. Mixing WT and HET does not seem completely appropriate.

We understand the concerns. We initially sequenced 3 independent, biological replicates for each genotype (WT, Het and KO). In the analysis presented in the initial manuscript, we were only showing the samples that were derived from the same gender of the embryos since we observed skewing/PCA clustering based on gender and not based on genotype. We have now performed corrections for gender bias across our samples and have used all the samples for the analysis.

Still, one of the KO samples is not good enough not be included due to the poor quality of the sample, so we only have 2 biological replicates for I κ B α KO.

After the gender correction, we first tested if there are significant differences in gene expression between WT and Het samples. We only see 8 genes (not belonging to specific pathways) differentially and significantly expressed between I κ B α WT and Het.

Bulk RNA seq- WT vs Het comparison

This result agrees with our phenotypic analysis (LSK, LT-HSC determination) where we don't detect any significant changes. To be more transparent, we now present the gene expression analysis as a heatmap depicting each sample and its gene expression level.

In this case, the KO samples cluster together whilst the WT and Het samples are intermixed. Moreover, the “AGM” HSCs related genes are consistently up regulated in both KO samples compared to the 5 WT/Het samples.

Reviewer #2 (Remarks to the Author):

Overall, the studies presented are interesting and address important questions: what is the genetic basis of functional Heterogeneity of HSCs, how are functional differences established and maintained during ontogeny, and what are the main similarities and differences between embryonic and adult regulation of HSC behavior? The main conclusion of interest is that β promotes a distinct behavior within a Heterogeneous population of HSCs, upstream of an NF κ B/inflammatory signaling-independent upregulation of RA-responsive genes to drive “dormancy” and the proposition that distinct RA-responsive cassettes mediate or partially mediate this behavior over the course of ontogeny.

Our main concerns revolve around 1) lack of robust establishment of phenotypes, 2) lack or robust establishment of putative mechanistic explanations, 3) overinterpretation of results and inadequate methods, repetitions, and parallel means to establish conclusions, 4) unclear writing and presentation of data, 5) inadequate description and documentation of methodology. We believe that with major revision and additional experiments, these findings could be suitable for Nature Cell Biology? and of high interest to its readers, assuming new results do not substantially alter the overall conclusions.

MAJOR

1. The putative physical mechanism of β inactivation of genes via PRC2 methylation and independent of NF κ B inhibition is not convincingly demonstrated, nor is the primary role of RA-responsive genes in this alleged phenotype.

We have now gathered further evidence that I κ B α can interact with PRC2 by performing Proximity Ligation assay (PLA) on E11.5 AGM sections for I κ B α and EZH2, a prominent member of the PRC2 complex.

Here, we find nuclear accumulation of interaction points (PLA signal) between EZH2 and I κ B α in IAHC, suggestive of physical interaction. Additionally, genes that are upregulated in I κ B α KO LT-HSCs enrich for EZH2 and EED by ChEA, indicating that the upregulated genes in I κ B α are possible targets of PRC2.

Finally, we detect I κ B α binding at the *rarg* promoter in LT-HSCs, and I κ B α KO LT-HSCs decreased levels of H3K27me3 mark (that is dependent on PRC2) in I κ B α E14.5 in the promoter of *rarg*.

All these findings indicate that I κ B α and PRC2 cooperate to regulate retinoic acid receptors.

Both retinoic acid receptors, *Rar α* and *Rarg*, have been established as critical regulators of dormancy and self-renewal in the adult bone marrow LT-HSC compartment (Cabeza-Wallscheid et al, 2017 and Schoeneberger et al, 2022, Purton et al, 2006 and 2008). Both these genes are significantly higher expressed in the I κ B α KO LT-HSCs compared to WT/Het. Moreover, when we compare dormancy signature (Cabeza-Wallscheid et al, 2017) with our I κ B α KO RNA seq, we see a good correlation.

In short, our molecular and functional analysis indicates that the LT-HSC population in I κ B α KO presents an elevated dormancy phenotype, and this is evident in the deregulation of dormancy related genes, including *Rar α* and *Rarg*.

2. The manuscript needs to be significantly edited for clarity and figures—especially schematics—need to be reworked to improve understandability.

We have extensively revised the manuscript for text and figures and hope it is easier to understand now.

3. We do not think examination of hematopoiesis in the P5/6 liver is meaningful.

Hematopoiesis has moved to the BM by then. If the authors believe this tissue and stage are meaningful, documentation is needed and a clearer exposition of relevance.

During the first week of life, murine hematopoiesis continues in the fetal liver, as well as BM (Yokomizo et al, 2022; Luis et al, 2012), this is the reason why at this time both organs should be analyzed.

4. Is looking at frequency of hematopoietic precursors meaningful? We believe that establishment of absolute numbers of HSPCs at stages examined needs to inform the percentage frequencies.

In the manuscript, we have determined the frequency of the LSK population from the lineage negative gate, or the frequency of LT-HSC from the LSK gate, as this is an established way to present FACS data. We agree that we did not clearly indicate from which gate(s) the determination was established and have now added this in the figure legends. Ultimately, we suppose that the reviewer is concerned if the total cellularity is significantly skewed between the genotypes. Considering this concern, we performed total cell number counts for WT, Het and KO E14.5 fetal liver and newborn liver and bone marrow.

In all tested hematopoietic sites, we did not detect gross differences between the genotypes. We therefore believe that showing the frequencies reflects the true differences between the genotypes.

5. Overall, the HSC defects in the absence of IκB shown in this manuscript are not convincing. For example, the authors show that there is a decrease in the percentage of AGM CD31+cKit+CD45+GFI1+EPCR+ HSCs, but the representative flow plot differences are predicated on the addition of GFI1 as a marker (Fig. S2B). If just EPCR is considered, it appears there may be a higher number CD31+cKit+CD45+EPCR+ HSPCs in the IκB KO. GFI1 expression in these populations has not been functionally shown to further purify HSCs and the papers referenced in the manuscript only show EPCR as a functional marker for HSCs, not GFI1. It is conceivable that there are simply lower levels of GFI1 in IκB embryos that do not correlate with HSC fate or function.

The Gfi1 reporter is a very robust HSC marker at the initial stage of HSC biology (Thambyrajah et al, 2016). Nevertheless, we determined the number of CD31+cKIT+CD45+Sca-EPCR high population in WT and KO which is now in the main figure. As already established using the GFI1 reporter, we detect less HSCs in the AGM defined as CD31+cKIT+CD45+SCA1+EPCR+.

Additionally, and to further validate the impact of I κ B α loss already in the first (pre-) HSCs, we have FACS purified endothelial cells (CD31+cKIT-) and IAHC (CD31+CKIT+) from E 11.5 AGMs to perform qPCRs for Rar α and Rar γ . We found significant over-expression of Rar γ in the cKIT population of the I κ B α KO AGMs.

Furthermore, the number of WT LSK and LT-HSC samples is too low and significant differences are only detected between HET and KO samples. The authors assume that there are no differences between WT and HET samples (which is based on a very low number of replicates) and therefore it is ok to compare HET and KO samples as if they were comparing WT and KO samples. Lack of Het phenotype needs to be established. This assumption has also been used for bulk RNA sequencing analyses, in which WT and HET HSPCs were pulled together.

We have now added more WT genotypes in the analysis for newborn liver and bone marrow LT-HSC determination. Overall, we have not detected any significant differences in LT-HSC between I κ B α WT and Het.

We do realize that there is a lower number of LSK in the bone marrow of newborns, which resulted from a technical problem. The samples were not depleted for lin- cells before the FACS analysis and therefore, there was a big variation in the gating. Now, we have performed the FACS analysis of new samples after lineage depletion. We have now only considered the new samples for the LSK determination.

Due to the reviewer's suggestion, we have now re-analysed our RNA seq data thoroughly again. We have corrected samples for gender bias and now added 2 more WT, and 1 more Het sample to the analysis. We now have enough samples to determine the molecular differences between I κ B α WT and KO LT-HSCs.

In the new PCA plots, the first principal component (PC1) which indicates the driving factor of difference between the samples, is now driven by genotypes. Most importantly, both WT and Het are distributed along the same horizontal axis compared to I κ B α KO, meaning that they are

more similar to each other than to Ikb α KO. In hierarchical clustering, both KO cluster together while the WT and Het samples are intermixed, further suggesting that they are indifferent.

Additionally, we performed DEG analysis comparing WT and Het samples.

Bulk RNA seq- WT vs Het comparison

We found only 8 genes that were significantly different between WT and HETs and these genes are not determinants of our phenotype described in the manuscript. In short, we have molecular and cellular evidence that WT and Hets are similar.

6. Fig. 2F putatively shows a “worsening” diminution of LT-HSCs over development from E11.5 to P6/7. No data is presented on the frequency of LSK-SLAM at E11.5 or P6/7, so not clear where these ratios came from. The legend says “taken from 2Eii,” but there are no E11.5 or P6/7 numbers in those charts.

At E11.5 the naïve HSC population does not express CD150/SLAM marker yet, this is also the reason why we need to use other markers such as Gfi1 or EPCR/Sca1 to determine the HSC population in the AGM. We apologize for the mislabeling of P5/6 as P6/7. Indeed, the ratios for the E14.5 and newborn liver and bone marrow are derived from the frequencies established in 2Eii and the frequencies for the E11.5 AGM is derived from Fig 2Dii. We apologize for this error and have now clarified it in the figure legend.

7. The exclusion of NF- κ B signaling is not convincingly demonstrated. Evidence put forth that NF- κ B pathway/inflammatory signaling is unaffected includes 1) Fig. S2B, which claims no elevation in p65 signal in ECs of I κ B α KO, but a) it looks like p65 might be up and b) IHC is not really an adequate approach to address alterations in p65. 2) Lack of elevated expression of NF- κ B target genes in WT vs. KO sequencing of E14.5 FL (Fig. S4B), but 7/12 genes show “significance” and several of them appear to show overt significance and possibly even a Het phenotype. A credible piece of evidence: GSEA analysis shows upregulation of non-NF- κ B pathways (S4D labeled “F”).

In our revised manuscript we refrain from the claim that this PRC2-I κ B α activity is entirely independent from NF- κ B activity. As shown in figure 1, and in the RNA sequencing data, we cannot totally exclude NF- κ B activity. Nevertheless, the phenotype we describe here, is specifically affecting the LT-HSC fraction, and is not driven by classical NF- κ B activity.

In the initial study describing the I κ B α KO phenotype, the authors also extracted embryonic fibroblasts from all genotypes of the I κ B α mouse line and assessed the levels of several NF- κ B proteins by Western blot. Importantly, they did not detect any difference in the levels of p65, c-Rel, p105, p50, p100 and I κ B β , nor in the nuclear p65 in basal conditions which remain cytoplasmic even in the absence of I κ B α (Beg et al, 1995; Klement et al, 1996). However, in one of the initial studies, hematopoietic cells were also examined and shown to have elevated NF- κ B signalling in some, but not all known NF- κ B targets (Keg et al, 1995).

As commented above, we have revised our RNA seq data. Here, we also repeated our analysis to test if NF- κ B pathway was over-represented in I κ B α KO. Whilst some of the NF- κ B targets are upregulated in the I κ B α KO genotype, a GSEA analysis against the KEGG annotated NF- κ B signature (mmu04064) did not result in a significant correlation. This indicates that there is a NF- κ B deregulation, but most likely not the driving factor of the phenotype in the LT-HSCs of I κ B α KO.

In addition, we have now sequenced the RNA of single LSKs from WT and KO E14.5 FL and we queried our data set for the same NF- κ B KEGG pathway. In none of the identified populations, there was any significant difference in the NF- κ B signature between WT and KO cells, indicating that NF- κ B pathway is not overactivated in the I κ B α KO HSC and progenitors.

8. The assertion that maturation of HSCs from an AGM-like gene signature to a FL-like gene signature—especially predicated on elevated expression of *Gpr56*, *Gata2*, *Neur13*, *Sox18*, and *Smad6*—needs better support. Experiments or references indicate that these genes are normally downregulated during this transition and that the “preserved” gene expression is similar to normal in the AGM. The statement, “this HSC pool is already compromised at the site of their origin in the E11.5 AGM since we find that the low number of $\text{I}\kappa\text{B}\alpha$ KO LT-HSC that are present in the FL at E14.5 retain embryonic/AGM specific signature” is moreover an overstatement. The authors need to present evidence that the problem is not due to failure of maturation in the FL.

To address these concerns by the reviewer, we have now first overlapped the top 300 genes expressed in AGM- HE/HSCs (Thambyrajah et al, 2023) and the E14.5 fetal liver (Manesia et al, 2017) in a Venn Diagram.

Only a small fraction of genes (1.9% accounting for 11 genes) is common between them suggesting that AGM derived HSCs have distinct molecular profiles. We then overlapped the list of genes deregulated in $\text{I}\kappa\text{B}\alpha$ KO with the list of genes distinct for AGM HSCs or fetal liver LT-HSCs. Comparing the DEG in $\text{I}\kappa\text{B}\alpha$ KO samples to their Fetal liver equivalent resulted in poor overlap and non-significant p-value (32 genes overlap but the p-value suggests this is not significant).

In comparison, we found 6.1% (101 genes in common between $\text{I}\kappa\text{B}\alpha$ KO LT-HSCs samples and AGM-HSCs with statistical significance.

The analysis indicates that a high number of AGM-HSC associated genes are retained in E14.5 $\text{I}\kappa\text{B}\alpha$ LT-HSCs. In the next step, we conducted GSEA analysis with the RNA seq samples from $\text{I}\kappa\text{B}\alpha$ WT/Het and $\text{I}\kappa\text{B}\alpha$ KO to further validate this finding.

The top 300 genes expressed in WT FL LT-HSCs does not enrich significantly for $\text{I}\kappa\text{B}\alpha$ WT or KO, but the gene set derived from AGM-HSC shows significant enrichment in the $\text{I}\kappa\text{B}\alpha$ KO samples. To demonstrate clearly which of the AGM-HSC genes are retained in the $\text{I}\kappa\text{B}\alpha$ KO LT-HSCs, we have now represented AGM-HSC gene set as a heatmap and plotted the expression levels of all E14.5 $\text{I}\kappa\text{B}\alpha$ LT-HSCs RNA seq samples (Please see Suppl table T2 for gene names). Moreover, green highlighted genes along the right bar in blue are significantly up-regulated in $\text{I}\kappa\text{B}\alpha$ KO LT-HSC samples. We still find *Gpr56*, *Gata2*, *Neur13* and *Smad6* significantly up-regulated (we have now not detected *Sox18* as significantly up-regulated).

The second concern the reviewer raises is the possibility that these HSCs fail to mature in the E14.5 fetal liver. We agree with his remark. We are demonstrating that these E14.5 Fetal liver HSCs retain AGM-like features. The cause for this phenotype is most likely their slow cycling/elevated dormancy program. In support of our hypothesis, now we have evidence including:

- We detect $\text{I}\kappa\text{B}\alpha$ in nuclei of IAHC (Figure 3D-F)
- We detect interaction between $\text{I}\kappa\text{B}\alpha$ and EZH2 in IAHC (Figure 4B)
- that the number of AGM-HSCs is already reduced (Figure 3Gii and Suppl Figure S5G)
- IAHC in $\text{I}\kappa\text{B}\alpha$ KO already overexpress $\text{Rar}\gamma$ (Figure 4G)

We therefore believe that the phenotype is already established in the AGM. Lastly, in transplantation assays (in a WT niche), these LT-HSCs still show a significant phenotype; first they lag in their donor contribution but over time outperform their WT counter parts.

9. The reliance on an unpublished AGM HSC (and HE) gene signature for the comparison demonstrating that $\text{I}\kappa\text{B}\alpha$ KO cells are more AGM-like is not really acceptable. The GSEA comparison parameters need to be better described and validated. Is comparing different size gene sets really meaningful?

We understand the reviewers' concerns. The data sets is now available in biorxiv (<https://www.biorxiv.org/content/10.1101/2023.04.19.537430v1>) and we have now equalized the number of genes in both data sets to the top 300 genes expressed.

10. No direct evidence that H3K27me3 alterations are directly because $\text{I}\kappa\text{B}\alpha$ /PRC2 defects in the KO HSCs. CUT & TAG assay shows that H3K27me3 alterations exist in the KO, but the alleged mechanism has not been shown in this situation. The occupancy of the $\text{Rar}\alpha$ and $\text{Rar}\gamma$ loci by $\text{I}\kappa\text{B}\alpha$ and its putative connection to "decreased methylation" in nearby regions of unknown significance in the gene bodies is not persuasive (Fig. 4A-C, Fig. S5B-C). The diminution in methylation is not particularly severe and of unknown functional consequence. If there is indeed decreased methylation, the dependence on $\text{I}\kappa\text{B}\alpha$ occupancy is unclear.

In the revised manuscript we have added several results and evidence that indicate PRC2- $\text{I}\kappa\text{B}\alpha$ interactions leading to the regulation of retinoic acid receptor expression levels.

In the first instance, the gene up regulated in $\text{I}\kappa\text{B}\alpha$ KO LT-HSCs include a significant number of genes that are targets of the PRC1 and PRC2 (ChEA enrichment for BMI1, EZH2 and EED) and $\text{RAR}\gamma$, indicating that genes that are up-regulated by the loss of $\text{I}\kappa\text{B}\alpha$ are also targets for PRC and $\text{RAR}\gamma$.

To provide evidence for the interactions between PRC2 (represented by EZH2) and $\text{I}\kappa\text{B}\alpha$ in IAHC we performed PLA experiments and detected interaction points in nuclei of IAHC. These findings strongly suggest a cooperation between PRC2 and $\text{I}\kappa\text{B}\alpha$.

Further evidence for the PRC2- $\text{I}\kappa\text{B}\alpha$ axis regulating retinoic acid signalling is provided by cut and tag assays. We find reproducible binding of $\text{I}\kappa\text{B}\alpha$ to the promoters of $\text{Rar}\alpha$ and $\text{Rar}\gamma$, both of which are significantly upregulated in $\text{I}\kappa\text{B}\alpha$ KO LT-HSCs. Finally, we find reduced PRC2 dependent H3K27me3 methylation for $\text{Rar}\gamma$ in LT-HSCs of $\text{I}\kappa\text{B}\alpha$ KO, and more broadly, when

we look at motifs enriched in regions that are more hypomethylated in I κ B α KO LT-HSCs, we also find retinoic acid motifs.

All these findings tie together to a model where PRC2-IKBA together regulate retinoic acid signaling at different levels in LT-HSCs.

11. Cell cycle connection to injected BrdU (Fig. 5A-C) needs citations and controls to establish the methodology, efficacy, and validated timing for the conclusions drawn. For example, the authors need to describe how they translated BrdU incorporation to number of cells in G0 and established that a 2-hour post injection labeling period (5A) is meaningful.

BrdU injection in vivo in embryos to analyze IAHC proliferation has been previously established in our lab (Porcheri et al, 2020) and in Zape et al, 2017, in which cell cycle phase lengths were established for EHT and IAHC. We have now extensively revised our experimental BrdU data for short-pulse labeling (see below) and for label retaining (pulse-chase).

For short pulse experiments, we did 2 hr chase with BrdU at the early fetal liver stage (E12.5) in order to test the BrdU incorporation efficiency (or cells in S phase) in the first HSCs from WT and KO. Now we represented the percentage of labelled cells in I κ B α WT and KO. We agree that we do not know the cell cycle state of the unlabeled (formerly incorrectly designated as G0) LT-HSCs.

Representative BrdU sample analysed 2 hours after BrdU injection.

As expected, in a short time between BrdU injection and analysis, most of the BrdU signal is seen in the S phase and not in the G1 stage when cells have undergone a full cycle to have incorporated BrdU. When we compare the BrdU positive cells in I κ B α WT and KO LT-HSC fraction, we detect a significant lower percentage of BrdU incorporation rate in I κ B α KO.

In further support of the BrdU labeling experiment and the slow cycling state of I κ B α LT-HSCs, we have now also included Ki67 staining for LSK sub-populations.

In agreement with our BrdU data, we find lower numbers of Ki67 positive cells in the LT and ST-HSC compartment of I κ B α KO E14.5 fetal livers.

We hope these two approaches that determine the cycling dynamics of I κ B α KO LT-HSCs further support our claims of a slow cycling LT-HSC pool.

For pulse-chase BrdU (5B), validation of quiescence/dormancy based on injection and BrdU label-retention following a 4-day chase needs to be demonstrated as meaningful.

In the pulse-chase experiment, we want to assess the percentage of cells that retain BrdU after 4 days of labelling. But the reviewer is correct that this data is only meaningful if we can achieve a similar level of labeling. For this reason, we label at E10.5-E11.5, to ensure the most labelling efficiency of HSC when they are being generated.

We now include this control experiment, whereby we inject BrdU on two consecutive days (E10/E11) and analyze the FL HSCs one day later, E12.5) to determine the maximum level of labeling HSCs during EHT. In these experiments, we were able to label 50-85% of the LT-HSC population, both in I κ B α WT and KO, indicating that that HSCs or their precursors from both genotypes can be labeled to comparable levels at the EHT stage.

We assessed how many of the labeled LT-HSCs keep the BrdU label over 4 days (from E10 to E14.5, Figure 5D), which would further indicate a slower cycling of these cells over the FL expansion time. We focused on the BrdU labeled cells that are in G1 since these would be the still quiescent cells, ie not cycling or cycled at lower levels to still retain BrdU after 4 days of BrdU incorporation. Here, we find more (BrdU) label retention in I κ B α KO LT-HSCs.

We have also labeled I κ B α WT and KO embryos at E10/11 with BrdU and analyzed the FL after a even longer chase period, at E16.5. In this experiment, we could not really detect a reliable population of BrdU positive cells, also challenging due to low level of HSCs in the I κ B α KO. Thus, we believe that most I κ B α KO LT-HSCs may not be totally quiescent, but cycle less frequently than WT LT-HSCs.

12. Although BrdU labeling experiments show significantly less BrdU incorporation in KO LT-HSCs, suggesting a more dormant phenotype, the functional results, in particular secondary transplantations are not convincing.

We have now performed transplantation experiments with the same number of purified LT-HSC (LSKCD48-CD150+) to avoid any advantage in cell number from WT and, even in these conditions, we still see some delay for the I κ B α KO LT-HSC to catch up to WT levels in primary transplantations (Figure 6A-E), as with the previously presented transplantation assay performed with the LSK cells (current suppl Figure S8A-D).

In the secondary transplantations, however, initially both genotypes are comparable, but I κ B α KO recipients show again significantly higher chimerism at 16w, with an increased tendency starting already at 2 months post transplantation.

In addition, if HSC dormancy is affected in the context of I κ B α deletion, it should be studied in the newborn bone marrow.

We agree with the reviewer that studying HSC dormancy in the I κ B α KO newborn (or in older animals) would be of great interest, and this is an on-going subject of investigation that requires conditional deletion of I κ B α . However the focus of this manuscript is on the AGM and fetal phenotype of the I κ B α KO and the presence of dormant cells during this period. I κ B α KO mice die 1 week after birth, and this is due to massive inflammation (a phenotype that is not found during development). For this reason, the functional studies of HSCs may be affected by NF- κ B signaling at this time and this is reason we have not included them in our study.

MINOR

1. Zhou reference duplicated.

We will delete one copy of the reference.

2. This sentence—"In its [nuclear I κ B α] absence, RA levels remain elevated and prevent LT-HSCs to be activated"—does not accurately reflect the authors' model. Don't they mean, expression of RA-responsive genes, and not RA per se?

Our data supports that PRC2-I κ B α shuts down Rar α or Rar γ , but also may affect its downstream targets, and its downstream targets to control the proliferation/quiescent ratio of HSCs

3. What is the meaning of the increased LSK frequency in the P5/6 BM? It's glossed as "comparable."

We do realize that there is a lower number of LSK in the bone marrow of newborns, which resulted from a technical problem. The samples were not depleted for lin⁻ cells before the FACS analysis and therefore, there was a big variation in the gating. Now, we have performed the FACS analysis of new samples after lineage depletion. We have now only considered the new samples for the LSK determination and detect no significant differences between genotypes.

4. Fig. S4D mislabeled as S4F.

Thank you for pointing out. We have extensively re-arranged the figures now and hope everything is labeled appropriately.

5. References needed for support of genes listed as "key embryonic (AGM) HSC-associated genes" including Smad6, Sox18, Gpr56, Gata2, and Neurl3? With the exception of Gata2 and possibly Gpr56, these are not the most obvious set of AGM HSC-associated genes. p. 7 top paragraph. What about Runx1? CD41? Gfi1? Gata2 not capitalized.

Smad6 (Lempereur et al, 2018), Sox18 (Ho Sun Jung et al, 2023) and Neurl3 (Hou et al, 2020; Fadluallah et al), are established markers of AGM HE/HSC and marker genes of this population, and not fetal liver stage (Zhou et al, 2016). We choose these to highlight since they are commonly associated with Hemogenic endothelium/EHT.

6. Gene/protein nomenclature, capitalization, italicization, spelling, and standardization editing needed throughout the text and figures.

We have thoroughly revised our nomenclature.

7. Gpr56 label typo Fig. 3C.

Thank you for pointing out. We have extensively re-arranged the figures now and hope everything is labeled appropriately.

8. FL data set from Manesia et al., needs better description in the text: how old? etc.

This data set is from 2015 and 2017 where E14.5 fetal liver and adult bone marrow HSC are compared along each other (Manesia et al, 2015 and 2017). The list of genes considered for this study are now included in Suppl table T3.

9. Color key needed for S4E.

Thank you for pointing out.

10. What is the authors' proposed mechanism explaining the presence of uniquely H3K27me3 peaks in the β KO HSCs?

We think that uniquely present H3K27me3 peaks in the WT chromatin may represent an decreased accessibility to this loci and likely less expression in WT (while more expression in KO), since H3K27me3 is associated with gene silencing. Our proposed mechanism is that Ikb α KO HSCs cannot properly methylate H3K27 in specific targets because Ikb α is required to direct the PRC2 complex to these genes (see model in Figure 7)

11. The schematic for Fig. 4A is very difficult to understand.

We have now revised Fig 4A scheme to Figure 4C-E.

12. In the Methods section for BrdU labelling/staining, what does the "pf" in "2mg pf BrdU" mean?

Sorry for the typo.

13. Example flow cytometry plots and unstained/FMO controls for the gating should be shown.

We have now added FMO controls and exemplary FACS plots in the Suppl Figures.

14. There is a significant increase between HET and KO LSKs in P5 and P6 BM that is not mentioned at all; instead, frequencies of FL, BM and newborn liver LSKs are described as comparable.

Thank you for this remark. The samples were not depleted for lin- cells before the FACS analysis and therefore, there was a big variation in the gating. Now, we have performed the FACS analysis of new samples after lineage depletion. We have now only considered the new samples for the LSK determination and detect no significant differences between genotypes.

15. Figure 5A does not show lower uptake of BrdU by KO LT-HSCs as stated in the text.

We meant that more of the Ikb α KO are in G0 so less BrdU labeling. However, we have now revised this experiment since reviewer rightly pointed out that we cannot determine the cell cycle stage with a 2-hour pulse. As explained in point 11, we have focused on the BrdU+ cells within the LT-HSC gate and can show that fewer LT-HSCs are labeled in Ikb α KO.

16. Figure 2 A, B – the number of embryos imaged should be added to the figure legend.

We have now included this info in the figure legend (a total number of 12 embryos were analysed).

Cabezas-Wallscheid, N., Buettner, F., Sommerkamp, P., Klimmeck, D., Ladel, L., Thalheimer, F.B., Pastor-Flores, D., Roma, L.P., Renders, S., Zeisberger, P., et al. (2017). Vitamin A-Retinoic Acid Signaling Regulates Hematopoietic Stem Cell Dormancy. *Cell* 169, 807–823 e19.

Jung HS, Suknuntha K, Kim YH, Liu P, Dettle ST, Sedzro DM, Smith PR, Thomson JA, Ong IM, Slukvin II. SOX18-enforced expression diverts hemogenic endothelium-derived progenitors from T towards NK lymphoid pathways. *iScience*. 2023 Apr 8;26(5):106621. doi: 10.1016/j.isci.2023.106621. PMID: 37250328; PMCID: PMC10214392.

Hou, S., Li, Z., Zheng, X. et al. Embryonic endothelial evolution towards first hematopoietic stem cells revealed by single-cell transcriptomic and functional analyses. *Cell Res* 30, 376–392 (2020).

Muhammad Zaki Hidayatullah Fadlullah, Wen Hao Neo, Michael Lie-a-ling, Roshana Thambyrajah, Rahima Patel, Renaud Mevel, Irène Aksoy, Nam Do Khoa, Pierre Savatier, Laura Fontenille, Syed Murtuza Baker, Magnus Rattray, Valerie Kouskoff, Georges Lacaud; Murine AGM single-cell profiling identifies a continuum of hemogenic endothelium differentiation marked by ACE. *Blood* 2022; 139 (3): 343–356.

Ganuz, M., Hall, T., Myers, J. et al. Murine foetal liver supports limited detectable expansion of life-long haematopoietic progenitors. *Nat Cell Biol* (2022).

Lempereur A, Canto PY, Richard C, Martin S, Thalgott J, Raymond K, Lebrin F, Drevon C, Jaffredo T. The TGF β pathway is a key player for the endothelial-to-hematopoietic transition in the embryonic aorta. *Dev Biol*. 2018 Feb 15;434(2):292-303.

Manesia, J.K., Franch, M., Tabas-Madrid, D., Nogales-Cadenas, R., Vanwelden, T., Van Den Bosch, E., Xu, Z., Pascual-Montano, A., Khurana, S., and Verfaillie, C.M. (2017). Distinct Molecular Signature of Murine Fetal Liver and Adult Hematopoietic Stem Cells Identify Novel Regulators of Hematopoietic Stem Cell Function. *Stem Cells Dev* 26, 573–584.

Manesia, Zhuofei Xu, Dorien Broekaert, Ruben Boon, Alex van Vliet, Guy Eelen, Thomas Vanwelden, Steve Stegen, Nick Van Gastel, Alberto Pascual-Montano, Sarah-Maria Fendt, Geert Carmeliet, Peter Carmeliet, Satish Khurana, Catherine M. Verfaillie, Highly proliferative primitive fetal liver hematopoietic stem cells are fueled by oxidative metabolic pathways, *Stem Cell Research*, Volume 15, Issue 3, 2015

Patel, S.H., Christodoulou, C., Weinreb, C., Yu, Q., da Rocha, E.L., Pepe-Mooney, B.J., Bowling, S., Li, L., Osorio, F.G., Daley, G.Q., et al. (2022). Lifelong multilineage contribution by embryonic-born blood progenitors. *Nature* 606, 747–753.

Porcheri, C., Golan, O., Calero-Nieto, F.J., Thambyrajah, R., Ruiz-Herguido, C., Wang, X., Catto, F., Guillen, Y., Sinha, R., Gonzalez, J., et al. (2020). Notch ligand Dll4 impairs cell recruitment to aortic clusters and limits blood stem cell generation. *EMBO J* 39, e104270.

Purton LE, Bernstein ID, Collins SJ. All-trans retinoic acid delays the differentiation of primitive hematopoietic precursors (lin-c-kit+Sca-1(+)) while enhancing the terminal maturation of committed granulocyte/monocyte progenitors. *Blood*. 1999 Jul 15;94(2):483-95.

Purton LE, Bernstein ID, Collins SJ. All-trans retinoic acid enhances the long-term repopulating activity of cultured hematopoietic stem cells. *Blood*. 2000 Jan 15;95(2):470-7.

Schönberger, K., Obier, N., Romero-Mulero, M.C., Cauchy, P., Mess, J., Pavlovich, P. V., Zhang, Y.W., Mitterer, M., Rettkowski, J., Lalioti, M.E., et al. (2022). Multilayer omics analysis reveals a non-classical retinoic acid signaling axis that regulates hematopoietic stem cell identity. *Cell Stem Cell* 29, 131–148.e10.

Yokomizo, T., Ideue, T., Morino-Koga, S., Tham, C.Y., Sato, T., Takeda, N., Kubota, Y., Kurokawa, M., Komatsu, N., Ogawa, M., et al. (2022). Independent origins of fetal liver haematopoietic stem and progenitor cells. *Nature* 609, 779–784.

Zape JP, Lizama CO, Cautivo KM, Zovein AC. Cell cycle dynamics and complement expression distinguishes mature haematopoietic subsets arising from hemogenic endothelium. *Cell Cycle*. 2017 Oct 2;16(19):1835-1847.

Zhou, F., Li, X., Wang, W., Zhu, P., Zhou, J., He, W., Ding, M., Xiong, F., Zheng, X., Li, Z., et al. (2016b). Tracing haematopoietic stem cell formation at single-cell resolution. *Nature* 533, 487–492.

Reviewers' Comments:

Reviewer #1:

Remarks to the Author:

Changes have not been highlighted in the revised version, likely because large portions of the text were amended.

I still find that Authors have not provided enough experimental evidences to support their conclusions and some fundamental controls are missing.

1) Regarding my previous comment on the Contribution to the observed phenotypes from potential non-cell autonomous effects potentially arising in I κ Ba germline deficient embryos: Looking into levels of p65, as authors have investigated, although informative it does not capture perturbations in downstream NF- κ B signaling pathway. Rather investigating downstream targets would be more informative.

2) To support conclusions on the specificity for the punctuated pIKBa immunofluorescence (IF) including IF in the IKBa KO embryos would have been highly desirable. Additionally, how was I κ Ba accumulation measured in Fig 3D? It is not clear in the text.

3) For the proximity ligation assay, no negative control is provided to demonstrate that the signal is specific.

4) I still find issues in the interpretation of the dormant HSC population in I κ Ba deficient embryos. In the previous version there were a number of concerns regarding gating of G0 versus G1 in BrdU plots. Now authors have provided FMO for BrdU although not for Ki67. Also for some experiments it is not clear how some populations were assigned. For example Fig 5E how did authors define G0? Which staining? FACS plots are needed for this experiment as critical for their conclusions. Similarly, FACS plots are required for Fig 5D as both of these experiments seem critical to their conclusions on slow cycling HSCs.

I still don't agree with the term "dormant" used by the authors as most LT-HSCs become BrdU+ after two BrdU injections. Slowly cycling seems more appropriate.

5) Regarding the interpretation of transplantation experiments on the dormancy of transplanted HSCs, it is also possible that this phenotype of increased engraftment post-tertiary transplant may not have much to do with the initial dormancy state of the HSCs and rather reflect aspects related more to adult KO HSCs. Authors do not discuss on this aspect. For transplantation studies, authors did not indicate if engraftment was multilineage or if any lineage bias was detected.

6) Even though authors indicated in their rebuttal letter that they have amended nomenclature for gene names (rara, cdk6...), most of them still remain in the text. They have neither eliminated the word "irrefutably" contrary to what they indicated in their letter.

Minor points

SF2a: it indicates E16.5 embryos? is this correct or is it E14.5?

SF2B: only 2 wt embryos.

Reviewer #3:

Remarks to the Author:

Thambyrajah et al. present a revised submission supporting their claim that I κ Ba plays a key role in regulating the dormancy of HSCs during early ontogeny in mouse. The revised version is improved

and many of our concerns have been addressed; however, there are still a number of issues for concern. Overall, we do believe the story and think the authors probably have the outlines generally correct, but there are number of technical issues we would like to see fixed. Especially: the statistical analyses are missing or in some cases not appropriate. Imaging experiments constitute major support for claims and there is little quantitation of those effects or indication of how representative or reproducible the key findings are. It also looks like the serial transplant experiments presented in Fig. 6 are either incorrectly described or there may have been major technical issues. We would like to see this research published, but with the issues outlined fixed.

Major comments

1. We appreciate that more replicates were added in figure 2B for the LSK and LT-HSC analysis of P5/P6 newborns and E14 FL, but it is not clear if there is statistical significance of the differences in phenotype between WT and KO. The following statement: "However, the number of LT-HSCs was significantly reduced in all hematopoietic sites of I κ Ba KO at newborn and FL stages (Figure 2Bii and 2Cii)" is an overstatement as there are no significant differences in LT-HSCs between I κ Ba WT vs KO. Bars with NS should be added to the LT-HSC graphs between WT and KO. The statistical analyses used should be added to the figure legend.
2. Overall, there seems to be a lack of rigor in the statistical analysis and reporting of statistical tests used for the different figures, and often, conclusions seem to be drawn from a single image without quantification. Statistical tests, n numbers and quantification methods should be added to the figure legends everywhere they are used.
3. All the t-tests used for statistical analysis are parametric, which means that all the data follow a normal distribution, which is very questionable. Many of the data do not look like they would pass a normality test, and therefore non-parametric tests should be considered in such instances. Could the authors verify the normality of the distribution and perform the correct statistical analyses?
4. The authors state that "In brief, we detect phosphorylated Ser^{32,36} I κ Ba in IAHC and find that the number of HSC precursors in the AGM is lower in I κ Ba KO. This difference in the number of LT-HSCs between WT and KO increases throughout hematopoietic development suggest that the HSCs population in I κ Ba KO does not expand at the same rate as their WT counterpart." Could this be due to a reduced specification of HSCs in the KO AGM rather than a reduced expansion?
5. We still do not understand the results presented in Fig. 4G. Do the results show increased abundance of Rarg transcript in IAHC I κ Ba KO cells vs. WT but not in endothelium by qRT-PCR? Why is this a violin plot? It seems likely that they authors are showing the collective elevation of transcripts of genes that are activated by RAR γ —that would provide stronger support for the model and be more in line with the data in Figs. D and F. Could the authors please clarify and if necessary update either the description and text or the presentation format?
6. The functional validation of dormancy is questionable. If 200 LT-HSCs alone were transplanted into primary recipients with no radioprotective dose of helper/competitor cells, how can the % of donor chimerism be so low in some mice both in primary and secondary transplants? More surprising is the massive decrease in donor chimerism in secondary recipients injected with WT cells. Is it possible that there was a technical problem? The use of 1 million whole bone marrow cells prevents us from making any conclusions in secondary and tertiary transplants, as the number of LT-HSCs compared to other bone marrow cells might be higher in the KO compared to WT BM. To determine whether there is a change in dormancy, the same number of LT-HSCs should have been sorted and transplanted into secondary and tertiary recipients.

Minor comments

1. Introduction – lines 8 and 9, "region" is repeated before and after AGM. Remove one of them.
2. Introduction - A citation (or more) should be added to the pre-HSC and T1 and T2 HSCs statements.
3. Introduction – in "upregulated upon on" – "on" should be removed.
4. "Inferring" needs to be changed to "implying."
5. A reference is needed for the statement, "Within the single cell RNA seq data set of the LSK, we assigned cell identity based on previous studies..."

Results

4. There is something wrong with the significance bars in Fig1 B.
5. It would be useful to have a description of how the different populations (from ECs to BM-adult HSCs) were defined either in the methods section or figure 1 legend.
6. Fig 1C shows GSEA plots of T1/T2 pre-HSC vs FL-HSC and ECs vs T1/T2 pre-HSCs but the text does not match this: "Ranked T1/T2 HSC vs EC and FL-HSC vs E12/14.5 fetal liver differentially expressed genes were correlated to the hallmark "Tnfa signalling via NF- κ B". The figure legend referring to 1C also needs to be updated.
It is also unclear what FL-HSC refers to, as the populations mentioned from this dataset are either FL_E12_HSC or FL_E14_HSC.
7. In supplementary figure S3A, the number of sections and embryos analyzed should be stated in the figure legend. Glancing at the images there seems to be an increase in p65 in the KO compared to the WT. It would be useful to have a quantification of p65+ cells between the 2 genotypes.
8. "by FACS to performed bulk RNA seq for in depth transcriptomic analysis of the samples" – removed at the end of performed.
9. Figure S4A and B – it is unclear what the figure shows. What are the different colors representing?
10. We think some of the figure callouts to Fig. S4A and B may be wrong in the text.
11. "We therefore examined the expression levels of key embryonic (AGM) HSC-associated genes such as Smad6, Sox18, Gpr56 (Adgrg1), Gata2 and Neurl3 in our FL data set and found higher expression levels of all tested AGM-HSC associated genes in the I κ Ba KO FL HSCs (Figure 3A and Suppl Figure 4B)" – there is a mis-reference to supplementary figure 4B.
12. "Importantly, classical NF- κ B activity was not substantially increased in the IAHC/HSC after loss of I κ Ba as assessed by IHC for p65 and p65 Ser536 (activated form of p65) (Suppl Fig S6A-C), which is in line with our finding in the FL (Suppl Figure S3C)." – there is a mis-reference to supplementary figure S3C, which shows adult BM and not FL.
13. Figure S6 A-C: "Importantly, classical NF- κ B activity was not substantially increased in the IAHC/HSC after loss of I κ Ba as assessed by IHC for p65 and p65 Ser536 (activated form of p65) (Suppl Fig S6A-C)" – there does not seem to be any quantification or statement of the number of sections/IAHCs analyzed. Could quantification and number of samples quantified be provided in order to support this conclusion?
14. It is difficult to discriminate between WT and HET in the heatmap on Figure S5A. Could more distinct colors be used?
15. It is unclear how Figure S5C supports the following statement: "This difference in the number of LT-HSCs between WT and KO increases throughout hematopoietic development (Suppl Fig S5C)"
16. Change "pre-cursors" to precursors.
17. In figure 6B and C, could you please match the colors of the legend to the graph?
18. The logic of the colony replating assay in Fig. 6F needs to be better explained.

REVIEWER COMMENTS

Reviewer #1 (Remarks to the Author):

Changes have not been highlighted in the revised version, likely because large portions of the text were amended.

I still find that Authors have not provided enough experimental evidences to support their conclusions and some fundamental controls are missing.

1) Regarding my previous comment on the Contribution to the observed phenotypes from potential non-cell autonomous effects potentially arising in I κ B α germline deficient embryos: Looking into levels of p65, as authors have investigated, although informative it does not capture perturbations in downstream NF- κ B signaling pathway. Rather investigating downstream targets would be more informative.

We agree with the reviewer that investigating downstream targets should be more informative than p65 levels. For this reason, we have investigated NF κ B signature in single cell Foetal liver LSK subpopulations from WT and KO embryos without finding any significant enrichment of this signature in different subpopulations of the LSK KO cells compared to WT (Fig 2E and Suppl Figure S4E). This signature is neither significantly enriched in bulk RNA sequencing from KO LT-HSCs compared to WT by GSEA (Figure 2G). However, functional analysis of overexpressed genes in I κ ba KO LT-HSCs do show enrichment of some NF κ B related functions, such as TNF α signaling or inflammatory response, suggesting that some NF κ B functions may be enhanced (Fig 2F). In summary, we find that some NF- κ B target genes may be deregulated in the I κ B α KO HSCs, however most of the NF κ B signature is not significantly enhanced in the absence of I κ B α .

This was included in the text in page 7:

Interestingly, genes that were over-expressed in I κ B α KO are genes involved in several inflammatory pathways, including "Tnf α signaling via NF κ B" (Figure 2F). Nevertheless, a GSEA for the KEGG gene set of NF- κ B (mmu04064) did not show a significant overall enrichment of this pathway in I κ B α KO LT-HSCs (Figure 2G). Moreover, we only found a few of the genes in this set significantly DEG in our RNA seq data set (Suppl Figure 5A). We reason that although elements of inflammatory signaling pathways are deregulated, this de-regulation is not the main transcriptional change occurring in I κ B α KO.

We agree that we cannot rule out non-cell autonomous effects contributing to the phenotype but based on our RNA seq and transplantation of purified E14.5 LT-HSCs, we do believe that we are predominately assessing the cell intrinsic contribution of I κ ba to HSC dormancy/low cycling at the specification/emergence stage.

2) To support conclusions on the specificity for the punctuated pI κ B α immunofluorescence (IF) including IF in the I κ B α KO embryos would have been highly desirable. Additionally, how was I κ B α accumulation measured in Fig 3D? It is not clear in the text.

The number of I κ ba accumulating cells/cluster were counted manually. We validated the specificity pI κ B α Ser32/36 and I κ ba (total) staining in the clusters using the *Gfi1:tomato* transgenic line as a marker for HE and IAHC. We performed IHC on lateral sections of GF11/I κ ba het and GF11/I κ ba KO AGM. These control stainings have now added as an additional panel to Suppl Figure S5D (for I κ ba) and a new Suppl Figure S5E for pI κ B α Ser32/36. Some unspecific staining in nucleosomes can be observed with pI κ B α Ser32/36 in all the KO cells, however, fine punctuated staining spread in the nucleus of cluster cells is specific.

3) For the proximity ligation assay, no negative control is provided to demonstrate that the signal is specific.

As a negative control, we omitted the primary EZH2 antibody in the PLA assay and assessed the PLA signal which is provided in Supp Figure S6G (former S6E).

4) I still find issues in the interpretation of the dormant HSC population in $I\kappa B\alpha$ deficient embryos. In the previous version there were a number of concerns regarding gating of G0 versus G1 in BrdU plots. Now authors have provided FMO for BrdU although not for Ki67. Also for some experiments it is not clear how some populations were assigned. For example Fig 5E how did authors defined G0? Which staining? FACS plots are needed for this experiment as critical for their conclusions. Similarly, FACS plots are required for Fig 5D as both of these experiments seem critical to their conclusions on slow cycling HSCs. I still don't agree with the term "dormant" used by the authors as most LT-HSCs become BrdU+ after two BrdU injections. Slowly cycling seems more appropriate.

We have now added a FMO control for Ki67 in Suppl Figure S7A. We used BrdU/ DAPI (Figure 5D) and Ki67/DAPI (Figure 5E) staining. Please note that we are plotting the percentage of BrdU+ (retaining) cells within LT-HSCs in Figure 5D. In Figure 5E, we designated G0 as cells that are not labelled by Ki67 and are 2n by DAPI. We have now included the FACS plots in the new Supps Figure S8 A and B.

We agree that this cell population could be also called slow cycling and we already refer to them as slow cycling in the manuscript, ie

Introduction: *In addition, the $I\kappa B\alpha$ KO LT-HSCs manifest a dormancy signature at the molecular level, and functionally, they display a **slow-cycling** and dormant behavior in Ki67 and BrdU labeling experiments.*

Results: *Interestingly, in the downregulated pathways, we find several processes, including *myc*-targets, Oxidative phosphorylation, G2M checkpoint and glycolysis in $I\kappa B\alpha$ deficient LT-HSCs (Figure 3C), reminiscent of quiescent HSCs and indicating a **slow cycling/dormant cell state**.*

Discussion: *In further support for a low cycling state, we uncovered a molecular dormancy signature in $I\kappa B\alpha$ KO LT-HSC that is akin to dormant bone marrow HSCs. Although, HSCs are reported to enter the dormant state around 3 weeks after birth (Bowie et al, 2006), we detect cells compatible with a dormant phenotype already in the AGM in the absence of $I\kappa B\alpha$. Our results thus suggest that a small population of HSCs may be acquiring the dormant phenotype already in the AGM that regulated by RA and it is amplified in the absence of $I\kappa B\alpha$. This agrees with latest reports supporting the existence of a **less proliferative HSC pool** generated in the AGM (Yokomizo et al, 2022; Ganuza et al, 2022).*

5) Regarding the interpretation of transplantation experiments on the dormancy of transplanted HSCs, it is also possible that this phenotype of increased engraftment post-tertiary transplant may not have much to do with the initial dormancy state of the HSCs and rather reflect aspects related more to adult KO HSCs. Authors do not discuss on this aspect. For transplantation studies, authors did not indicate if engraftment was multilineage or if any lineage bias was detected.

The *Ikba* KO HSCs retain AGM-associated gene signature (Figure 3 A and B) hence we interpret that they are not more mature than their WT counterparts, but more immature.

In Suppl Figure S9E (former S8D) we do present the multi-lineage contribution of the transplanted *Ikba* WT and KO cells in primary and tertiary recipients. We have highlighted this in the text now as well.

Moreover, all blood lineages were reconstituted by both *IkBa* WT and KO LT-HSCs or LSK cells in primary and secondary recipients (Suppl Fig S9E).

6) Even though authors indicated in their rebuttal letter that they have amended nomenclature for gene names (*rara*, *cdk6*...), most of them still remain in the text. They have neither eliminated the word “irrefutably” contrary to what they indicated in their letter.

We apologise for these mistakes. We have now replaced “irrefutable” with “conclusively” in the manuscript and highlighted this change in yellow. We have also revised the manuscript for the correct nomenclature.

Minor points

SF2a: it indicates E16.5 embryos? is this correct or is it E14.5?

Yes, E16.5 as the pics and weight.

SF2B: only 2 wt embryos.

We do apologise for the low number of WT Newborn in this assay but, WT embryos are also only born in 1:4 ratio like the *Ikba* KO. Since the WT and HETs show similar levels of each blood lineage, and *Ikba* KO seems to show a mild, but consistent increase in Ter119 (Suppl Fig 1C), we did not examine more litters.

Reviewer #3 (Remarks to the Author):

Thambyrajah et al. present a revised submission supporting their claim that *IkBa* plays a key role in regulating the dormancy of HSCs during early ontogeny in mouse. The revised version is improved and many of our concerns have been addressed; however, there are still a number of issues for concern. Overall, we do believe the story and think the authors probably have the outlines generally correct, but there are number of technical issues we would like to see fixed. Especially: the statistical analyses are missing or in some cases not appropriate. Imaging experiments constitute major support for claims and there is little quantitation of those effects or indication of how representative or reproducible the key findings are. It also looks like the serial transplant experiments presented in Fig. 6 are either incorrectly described or there may have been major technical issues. We would like to see this research published, but with the issues outlined fixed.

Major comments

1. We appreciate that more replicates were added in figure 2B for the LSK and LT-HSC analysis of P5/P6 newborns and E14 FL, but it is not clear if there is statistical significance of the differences in phenotype between WT and KO. The following statement: “However, the number of LT-HSCs was significantly reduced in all hematopoietic sites of I κ B α KO at newborn and FL stages (Figure 2Bii and 2Cii)” is an overstatement as there are no significant differences in LT-HSCs between I κ B α WT vs KO. Bars with NS should be added to the LT-HSC graphs between WT and KO. The statistical analyses used should be added to the figure legend.

We have now carefully revised the manuscript and figure legends and Figures and added error bars and specified the type of statistical test used in the figure legend wherever required.

Additionally, we have retrieved more replicates in all Figure 2B panels since some more data had been collected. Importantly, following reviewer’s suggestion in point 3 and applying normality test over one-way ANOVA model residuals, only Figure 2Bii and Figure 2Cii comply with the normality assumption. Since the rest of the panels does not comply this assumption, a non-parametric test (Kruskal-Wallis and Dunn’s post-hoc test) was applied for all cases in Figures 2B and 2C for homogeneity and clarity for the reader. With this new analysis and data, the reviewer is correct that there is no significance in the LT-HSCs percentage remaining in the new born FL, while there is a statistically significant difference (p -value <0.05) in the LT-HSCs from newborn BM and E14.5 FL when comparing WT or HET against KO.

To conclude, this statement has been modified by removing “all” in the reviewed manuscript to:

“However, the number of LT-HSCs was significantly reduced in the hematopoietic sites of I κ B α KO at newborn and FL stages (Figure 2Bii and 2Cii and Suppl Figure S2A and B).

2. Overall, there seems to be a lack of rigor in the statistical analysis and reporting of statistical tests used for the different figures, and often, conclusions seem to be drawn from a single image without quantification. Statistical tests, n numbers and quantification methods should be added to the figure legends everywhere they are used.

We understand the reviewer’s concern, we have revised all the statistical analysis with a specialist and applied the most appropriate test. In point 3, we include the general statistical criteria applied in the reviewed manuscript.

We apologise for not mentioning the number of sections/embryos tested for each staining. We have now added this information in the figure legends, and in fact, we now show more cases in the Supplementary figure S3B and please also see answer to point 13 below.

3. All the t-tests used for statistical analysis are parametric, which means that all the data follow a normal distribution, which is very questionable. Many of the data do not look like they would pass a normality test, and therefore non-parametric tests should be considered in such instances. Could the authors verify the normality of the distribution and perform the correct statistical analyses?

We have conducted a thorough review of all the statistical analysis included in the paper and shown in the different figures. Based on the reviewer’s suggestion, we have applied the following criteria in the reviewed manuscript. Shapiro-Wilk test was used in all cases for verifying normality distribution.

In those cases where more than two groups were to be tested (i.e. WT, HET and KO), we tested the normality assumption over the one-way ANOVA residuals:

If there was no evidence to reject the null hypothesis (p -value > 0.05), an one-way ANOVA was applied.

On the contrary, a non-parametric test was chosen, specifically, Kruskal-Wallis test.

For previous scenarios (more than two groups), if ANOVA or Kruskal-Wallis test results was below significance level (p -value < 0.05), all pairwise comparisons were conducted with:

Dunn's test was a post-hoc test.

t-test as a post-hoc test. *This was the criteria to be applied but there was no case in this situation in the whole paper.

In those cases where there were only two groups to be tested (i.e. WT and KO), we tested the normality assumption over the raw data per group only if there were, at least, 4 values to be tested since normality violations are difficult to detect due to the power reduction. In those cases, with $n < 4$, a non-parametric test was applied. For the rest of cases:

If there was no evidence to reject the null hypothesis, then a t-test with Welch correction (to not assume equal variances) was applied.

On the contrary, a non-parametric test was chosen, specifically, a Mann-Whitney U test (Wilcoxon rank sum test).

All tests considered in the manuscript are unpaired except for data in Figure 5E for the comparisons between DMSO and Ro-41 treatment.

All tests were performed one-sided except for (i) those cases where Dunn's test was used (two-tailed) and (ii) Supplementary Figure 9D.

Finally, in Figures 3G and 6B, a multiple linear regression model was fitted with two covariates: Experiment (three different experiments to generate the data) and Group (WT or KO) for testing WT against KO controlling by experiment variability. The p -value reported is the result of applying a t-test over the estimated model coefficient for the Group variable ($\beta \neq 0$).

4. The authors state that "In brief, we detect phosphorylated Ser32,36 I κ B α in IAHC and find that the number of HSC precursors in the AGM is lower in I κ B α KO. This difference in the number of LT-HSCs between WT and KO increases throughout hematopoietic development suggest that the HSCs population in I κ B α KO does not expand at the same rate as their WT counterpart." Could this be due to a reduced specification of HSCs in the KO AGM rather than a reduced expansion?

This is a very interesting point, and we thank the reviewer for raising this question. Indeed, it is possible that the HSC numbers is already reduced at the specification stage in the AGM. We have now included this now in the Discussion.

"The low number of I κ B α KO LT-HSC that are present in the FL at E14.5 retain an embryonic/AGM specific signature, including Sox18, Gpr56 or Gata2, indicating impaired maturation and low proliferation, however we cannot exclude that HE specification may also be compromised."

5. We still do not understand the results presented in Fig. 4G. Do the results show increased

abundance of Rarg transcript in IAHC IκBa KO cells vs. WT but not in endothelium by qRT-PCR?
That's correct.

Why is this a violin plot? It seems likely that they authors are showing the collective elevation of transcripts of genes that are activated by RARg—that would provide stronger support for the model and be more in line with the data in Figs. D and F. Could the authors please clarify and if necessary update either the description and text or the presentation format?

By qPCRs from either CD31+cKIT- (endo) and CD31+cKIT+ (IAHC) we find increased levels of rarg only in IAHC but not in endothelial (endo) cells. We agree with the reviewer that violin plots are not appropriate for this representation and have now used bar charts instead.

In addition, as the reviewer says, we find that RARg targets are enriched in KO vs WT HSC transcripts as shown in the heatmap (Fig 4D), and additionally genes with RARg motif are in a more repressed state marked by H3K27 in the WT (Fig.4F)

6. The functional validation of dormancy is questionable. If 200 LT-HSCs alone were transplanted into primary recipients with no radioprotective dose of helper/competitor cells, how can the % of donor chimerism be so low in some mice both in primary and secondary transplants? More surprising is the massive decrease in donor chimerism in secondary recipients injected with WT cells. Is it possible that there was a technical problem? The use of 1 million whole bone marrow cells prevents us from making any conclusions in secondary and tertiary transplants, as the number of LT-HSCs compared to other bone marrow cells might be higher in the KO compared to WT BM. To determine whether there is a change in dormancy, the same number of LT-HSCs should have been sorted and transplanted into secondary and tertiary recipients.

We apologize for not clearly state that indeed radioprotective competitor cells were transplanted along with 200 LT-HSCs (as described in Materials and Methods). We have now added this info to the scheme (Figure 6A and Suppl Figure S9B) and in the manuscript:

"We therefore tested the hematopoietic activity of IκBa KO LT-HSCs (200 per recipient) or LSK (1000 cells) in competition with 500.000 lin- depleted bone marrow WT cells, respectively, in transplantation assay (Fig 6A)."

However, the percent of donor chimerism in primary recipients is almost 100% in most cases by 16 weeks (Fig 6B). In the secondary FL transplants, there is a drop in the percentage of chimerism as previously shown for FL cells (Ganuza et al, 2022), but our results show that the IκB KO cells can contribute to higher chimerism than WT in these conditions (Fig 6C).

Minor comments

1. Introduction – lines 8 and 9, "region" is repeated before and after AGM. Remove one of them.

Thanks, done.

2. Introduction - A citation (or more) should be added to the pre-HSC and T1 and T2 HSCs statements.

We have now added the citation of the study describing this process of HSC maturation.

"Some of the IAHC that are cKIT and CD41 positive (Pre-HSCs/T1 HSCs), gain HSC specific markers, including CD45, Sca1 and EPCR (CD201/PROCR) (T2 HSCs) (Zhou et al, 2016)."

3. Introduction – in “upregulated upon on” – “on” should be removed.

Thanks, done.

4. “Inferring” needs to be changed to “implying.”

Yes, that is correct. Thank you.

5. A reference is needed for the statement, “Within the single cell RNA seq data set of the LSK, we assigned cell identity based on previous studies...”

Thanks, we have now added the appropriate citation.

Within the single cell RNA seq data set of the LSK, we assigned cell identity based on previous studies (Suppl Figure S4 A-C, Suppl table T2) (Rodriguez-Fraticelli et al, 2018).

Results

4. There is something wrong with the significance bars in Fig1 B.

We thank the reviewer for the remark. Indeed, one of the error bars was duplicated. We have now corrected this.

5. It would be useful to have a description of how the different populations (from ECs to BM-adult HSCs) were defined either in the methods section or figure 1 legend.

Thank you for this suggestion. We have now added this information in the figure legend.

Figure 1: NF-κB signaling is enriched in AGM derived HSCs (A) UMAP projection colored by scRNAseq clusters where different HSC populations are highlighted. Single cell RNA sequencing data derived from Zhou et al, 2016. EC= CD31+ Cdh5+CD41-CD43-CD45-; T1 Pre-HSC= CD31+cKIT+CD41^{low}CD201^{high}CD45-; T2 Pre-HSC= CD31+cKIT+CD201^{high}CD45+; E12 FL-HSC= lin-Sca+CD201^{high}Mac1^{low}; E14 FL-HSC= CD45+CD201CD48-CD150+; adult BM-HSC= lin-Sca+cKIT+CD135-CD34- CD48-CD150+

6. Fig 1C shows GSEA plots of T1/T2 pre-HSC vs FL-HSC and ECs vs T1/T2 pre-HSCs but the text does not match this: “Ranked T1/T2 HSC vs EC and FL-HSC vs E12/14.5 fetal liver differentially expressed genes were correlated to the hallmark “Tnfa signalling via NF-κB”. The figure legend referring to 1C also needs to be updated. It is also unclear what FL-HSC refers to, as the populations mentioned from this dataset are either FL_E12_HSC or FL_E14_HSC.

We apologise. Indeed, this sentence is confusing. We have now corrected it to:

Ranked T1/T2 HSC vs E12/14.5 fetal liver (FL-HSC) and EC vs T1/T2 HSC differentially expressed genes correlated to the hallmark “Tnfa signalling via NF-κB” (Suppl table T1).

7. In supplementary figure S3A, the number of sections and embryos analysed should be stated in the figure legend. Glancing at the images there seems to be an increase in p65 in the KO compared to the WT. It would be useful to have a quantification of p65+ cells between the 2 genotypes.

For p65, we analysed 4 sections in total from two different embryos for each genotype. We have now added these numbers in the figure legend.

We also have quantified p65 nuclear localization between *Ikbα* WT and KO as an indication of p65 activation. We manually counted the number of cells with nuclear p65 accumulation in 4 different fields from each genotype, and if anything, we find increased number of nuclear p65 in *Ikbα* WT. We have now included this data in Figure S3B and C.

8. “by FACS to performed bulk RNA seq for in depth transcriptomic analysis of the samples” – remove d at the end of performed.

We have now corrected this, thank you.

9. Figure S4A and B – it is unclear what the figure shows. What are the different colors representing?

These plots are the basis of the pseudo time analysis. The pink colour in S4A-C depicts the most naïve cells, ie LT-HSCs. The blue cells are the MPPs (S4C). In S4A, the marker genes that distinguish LT-HSCs from the more committed cells are plotted (Rodriguez-Fraticelli et al, 2018). In S4B, the alterations in the population identifying genes are used to order the cells in a pseudo time line, ie their differentiation towards progenitor cells.

We have now extended the sentence in the manuscript to make it clearer.

Within the single cell RNA seq data set of the LSK, we assigned cell identity based on previous studies (Suppl Figure S4 A-C, Suppl table T2) (Rodriguez-Fraticelli et al, 2018) and confirmed the cell identity assignment by ordering the cells in a pseudo time trajectory based on their overall transcriptomic profile (Suppl Figure S4A-C).

10. We think some of the figure callouts to Fig. S4A and B may be wrong in the text.

The Figure S4A-C belong together as one entity since all three aspects of the analysis are dependent on each other. Please see above for a more detailed explanation.

11. “We therefore examined the expression levels of key embryonic (AGM) HSC-associated genes such as Smad6, Sox18, Gpr56 (Adgrg1), Gata2 and Neurl3 in our FL data set and found higher expression levels of all tested AGM-HSC associated genes in the I κ B α KO FL HSCs (Figure 3A and Suppl Figure 5B)” – there is a mis-reference to supplementary figure 4B.

This is true, we were referring to Suppl Figure 5B and have now corrected it in the manuscript.

12. “Importantly, classical NF- κ B activity was not substantially increased in the IAHC/HSC after loss of I κ B α as assessed by IHC for p65 and p65 Ser536 (activated form of p65) (Suppl Fig S6A-C) which is in line with our finding in the FL (Suppl Figure S3C).” – there is a mis-reference to supplementary figure S3C, which shows adult BM and not FL.

Thank you, in fact, we have now deleted this part of the sentence from the manuscript since we do see an increase in p65 nuclear localisation in I κ ba KO E14.5 FL.

13. Figure S6 A-C: “Importantly, classical NF- κ B activity was not substantially increased in the IAHC/HSC after loss of I κ B α as assessed by IHC for p65 and p65 Ser536 (activated form of p65) (Suppl Fig S6A-C)” – there does not seem to be any quantification or statement of the number of sections/IAHCs analyzed. Could quantification and number of samples quantified be provided in order to support this conclusion?

We have now stated in the figure legends the number of sections analysed for each staining.

For pP65Ser536, we analysed 4 AGM sections from I κ ba WT and 6 sections from I κ ba KO in 2 independent experiments. IAHC was identified with cKIT staining (and corresponding cells highlighted with yellow circle)

We have now quantified the mean fluorescence in the AGM of I κ ba WT and KO with ImageJ software.

We did not detect significant differences between Ikba WT and KO. We have now included some of this data as Supplementary Figure 6E in the manuscript.

14. It is difficult to discriminate between WT and HET in the heatmap on Figure S5A. Could more distinct colors be used?

Changing the colour in this particular graph would need to be followed by correcting the colour scheme in all the related graphs. We do not see the importance of distinguishing between WT and Hets since both genotypes are not significantly different.

15. It is unclear how Figure S5C supports the following statement: “This difference in the number of LT-HSCs between WT and KO increases throughout hematopoietic development (Suppl Fig S5C)”

We thank the reviewer for raising this concern. Indeed, the comparison of the HSC pools at different development times and niches was inappropriate. We have now plotted the HSC/LT-HSC determination at each time point for each genotype. The HSC pool in the Ikba WT increases over time compared to the IkBa KO HSC pool.

16. Change “pre-cursors” to precursors.

We have now changed this, thank you.

17. In figure 6B and C, could you please match the colors of the legend to the graph?

We have corrected this now.

18. The logic of the colony replating assay in Fig. 6F needs to be better explained.

We have now added this sentence to better explain the rationale.

LT-HSCs that are in a dormant/slow cycling state need more time to proliferate than their already activated counterparts (Cabezas-Wallscheid et al., 2017).

We have also decided to move Fig 6F to supplemental since it does not show a clear result and it is not easy to interpret. After revising the statistics and applying unpaired t-test, the difference between WT and KO in small colonies does not reach significance, although number of small KO colonies are higher in each independent experiment, variation between experiments is high. We have also added a new graph comparing each colony classification as Suppl Figure S9Biii and hope that the reviewers appreciate that overall, there is a higher proliferation rate in colonies derived from *Ikba* KO LT-HSCs. Since we have not been able to perform more experiments due to animal facility problems, we believe that supplemental is a better place for it. In the manuscript, we have changed "significant increase" to "showing an trend towards increase in *IkBα* KO".

References

Ganuza M, Hall T, Myers J, Nevitt C, Sánchez-Lanzas R, Chabot A, Ding J, Kooienga E, Caprio C, Finkelstein D, Kang G, Obeng E, McKinney-Freeman S. Murine foetal liver supports limited detectable expansion of life-long haematopoietic progenitors. *Nat Cell Biol.* 2022 Oct;24(10):1475-1486. doi: 10.1038/s41556-022-00999-5. Epub 2022 Oct 6. PMID: 36202972; PMCID: PMC10026622.

Reviewers' Comments:

Reviewer #1:

Remarks to the Author:

In this second revision authors addressed most of my comments.

1) Still, in my very initial report I indicated:

"In abstract and introduction, it seems that authors will be performing a deep study on the differences among HSCs and Hematopoietic Progenitor Cells (HPCs) based on recent findings indicating their potential independent embryonic origin. But authors just reported that they are able to find differences in HSC frequencies and not in HSPC frequencies in the embryo without going into any further investigation on this per se."

Still HPC are not the focus of their research, so it would be helpful to modify the Abstract as it is misleading:

"Recent findings support that Hematopoietic Stem Cells (HSCs) originate independently from Hematopoietic Progenitor Cells (HPCs) in the embryonic aorta-gonad mesonephros (AGM) region and thus, challenge previous studies on the dynamic evolution of these populations during development. In this new model, HSCs and HPCs are generated simultaneously with HSCs retaining a low proliferation index whilst HPCs expand extensively. It is unclear how the different proliferation rates between HPCs and HSCs are controlled".

2) In both my two previous reviews I highlighted that nomenclature for genes and protein names is quite poorly followed. Although they claimed in both revisions that they have reviewed and amended all this, there are quite a number of typos. For instance, nomenclature for RAR and RXR seems randomly used between lines 312 and 320 (rar, RXR, RxR...)

Reviewer #3:

Remarks to the Author:

Thambyrajah et al. present a revised submission supporting their claim that IκBα plays a key role in regulating the dormancy of HSCs during early ontogeny in mouse. We appreciate that the authors were highly responsive to our previous suggestions and we believe the manuscript is now suitable for publication with minor revision. We suggest revision for English spelling, grammar, usage, consistency in hyphenation (e.g. "over-expression" vs. "overexpression," "de-regulation" vs. "deregulation," etc.), and spacing between figure and number (e.g. "Figure2D" vs. "Figure 3G"). Overall we think this story is interesting and provides novel insight into the ontogeny and regulation of HSC emergence during development.

Reviewer #1 (Remarks to the Author):

In this second revision authors addressed most of my comments.

1) Still, in my very initial report I indicated:

“In abstract and introduction, it seems that authors will be performing a deep study on the differences among HSCs and Hematopoietic Progenitor Cells (HPCs) based on recent findings indicating their potential independent embryonic origin. But authors just reported that they are able to find differences in HSC frequencies and not in HSPC frequencies in the embryo without going into any further investigation on this per se.”

Still HPC are not the focus of their research, so it would be helpful to modify the Abstract as it is misleading:

“Recent findings support that Hematopoietic Stem Cells (HSCs) originate independently from Hematopoietic Progenitor Cells (HPCs) in the embryonic aorta-gonad mesonephros (AGM) region and thus, challenge previous studies on the dynamic evolution of these populations during development. In this new model, HSCs and HPCs are generated simultaneously with HSCs retaining a low proliferation index whilst HPCs expand extensively. It is unclear how the different proliferation rates between HPCs and HSCs are controlled“.

We are happy to hear that we were able to clear most of the concerns that the reviewer had. We have now also re-focused the abstract on HSC biology and hope that the reviewers agrees with the following version.

*“Recent findings suggest that Hematopoietic Stem Cells (HSC) and progenitors arise simultaneously and independently from each other already in the embryonic aorta-gonad mesonephros region, but it is still unknown how their different features are established. Here, we uncover *IκBα* (*Nfκbia*, the inhibitor of *NF-κB*) as a critical regulator of HSC proliferation throughout development. *IκBα* balances retinoic acid signaling levels together with the epigenetic silencer, *PRC2*, specifically in HSCs. Loss of *IκBα* decreases proliferation of HSC and induces a dormancy related gene expression signature instead. Also, *IκBα* deficient HSCs respond with superior activation to in vitro culture and in serial transplantation. At the molecular level, chromatin regions harboring binding motifs for retinoic acid signaling are hypo-methylated for the *PRC2* dependent *H3K27me3* mark in *IκBα* deficient HSCs. Overall, we show that the proliferation index in the developing HSCs is regulated by a *IκBα* - *PRC2* axis which controls retinoic acid signaling.”*

2) In both my two previous reviews I highlighted that nomenclature for genes and protein names is quite poorly followed. Although they claimed in both revisions that they have reviewed and amended all this, there are quite a number of typos. For instance, nomenclature for RAR and RXR seems randomly used between lines 312 and 320 (rar, RXR, RxR...)

We are very sorry that we missed some of the typos. We have once again thoroughly checked for nomenclature and hope that we rectified all of them.

Reviewer #3 (Remarks to the Author):

Thambyrajah et al. present a revised submission supporting their claim that IκBα plays a key role in regulating the dormancy of HSCs during early ontogeny in mouse. We appreciate that the authors were highly responsive to our previous suggestions and we believe the manuscript is now suitable for publication with minor revision. We suggest revision for English spelling, grammar, usage, consistency in hyphenation (e.g. “over-expression” vs. “overexpression,” “de-regulation” vs. “deregulation,” etc.), and spacing between figure and number (e.g. “Figure2D” vs. “Figure 3G”). Overall we think this story is interesting and provides novel insight into the ontogeny and regulation of HSC emergence during development.

We thank the reviewer for the positive comments. We are delighted that we could address all the concerns appropriately. We have again revised our manuscript for typos, nomenclature and inconsistencies.